



**Decreasing Trends of Particle Number and Black Carbon**
**Mass Concentrations at 16 Observational Sites in Germany**
**from 2009 to 2018**
Jia Sun[1], Wolfram Birmili[1,2], Markus Hermann[1], Thomas Tuch[1], Kay Weinhold[1], Maik
Merkel[1], Fabian Rasch[1,a], Thomas Müller[1], Alexander Schladitz[3,b], Susanne Bastian[3], Gunter
Löschau[3], Josef Cyrys[4,5], Jianwei Gu[4,5,c], Harald Flentje[6], Björn Briel[6], Christoph Asbach[7],
Heinz Kaminski[7], Ludwig Ries[2], Ralf Sohmer[2], Holger Gerwig[2], Klaus Wirtz[2], Frank
Meinhardt[2], Andreas Schwerin[2], Olaf Bath[2], Nan Ma[8,1], Alfred Wiedensohler[1]
[1]Leibniz Institute for Tropospheric Research (TROPOS), Leipzig, Germany
[2]German Environment Agency (UBA), Dessau-Roßlau, Germany
[3]Saxon State Office for Environment, Agriculture and Geology (LfULG), Dresden, Germany
[4]Helmholtz Zentrum München (HMGU), Institute of Epidemiology II, Neuherberg, Germany
[5]University of Augsburg(UA), Wissenschaftszentrum Umwelt, Augsburg, Germany
[6]Deutscher Wetterdienst (DWD), Meteorologisches Observatorium Hohenpeißenberg, Germany
[7]Institute of Energy and Environmental Technology (IUTA), Duisburg, Germany
[8]Institute for Environmental and Climate Research, Jinan University, Guangzhou, Guangdong 511443, China
[a]now at: Bundesanstalt für Materialforschung und -prüfung (BA), Richard-Willstätter-Str. 11, 12489 Berlin,
Germany
[b]now at: SICK Engineering GmbH, Ottendorf-Okrilla, Germany
[c]now at: Fraunhofer Wilhelm-Klauditz-Institut (WKI), Braunschweig, Germany
*Correspondence to*: Nan Ma (nan.ma@jnu.edu.cn) and Alfred Wiedensohler (alfred.wiedensohler@tropos.de)
**Abstract.** Anthropogenic emissions are a dominant contributor to air pollution. Consequently, mitigation
policies have attempted to reduce anthropogenic pollution emissions in Europe since the 1990s. To evaluate the
effectiveness of these mitigation policies, the German Ultrafine Aerosol Network (GUAN) was established in
2008, focusing on black carbon and sub-micrometer aerosol particles, especially ultrafine particles. In this
investigation, trends of the size-resolved particle number concentrations (PNC) and the equivalent black carbon
(eBC) mass concentration over a 10-year period (2009-2018) were evaluated for 16 observational sites for
different environments among GUAN. The trend analysis was done for both, the full-length time series and on
subsets of the time series in order to test the reliability of the results. The results show generally decreasing
trends of both, the PNCs for all size ranges as well as eBC mass concentrations in all environments, except PNC
in 10-30 nm at regional background and mountain sites. The annual slope of the eBC mass concentration varies
between -7.7 % and -1.8 % per year. The slopes of the PNCs varies from -6.3 % to 2.7 %, -7.0 % to -2.0 %, and -
9.5 % to -1.5 % per year (only significant trends) for 10-30 nm, 30-200 nm, and 200-800 nm particle diameter,
respectively. The regional Mann-Kendall test yielded regional-scale trends of eBC mass concentration, $N_{[30-200]}$
and $N_{[200-800]}$ of -3.8 %, -2.0 % and -2.4 %, respectively, indicating an overall decreasing trend for eBC mass
concentration and sub-micrometer PNC (except $N_{[10-30]}$) all over Germany. The most significant decrease was
observed on working days and during daytime in urban areas, which implies a strong evidence of reduced
anthropogenic emissions. For the seasonal trends, stronger reductions were observed in winter. Possible reasons
for this reduction can be the increased average ambient temperatures and wind speed in winter, which resulted in
less domestic heating and stronger dilution. In addition, decreased precipitation in summer also diminishes the



decrease of the PNCs and eBC mass concentration. For the period of interest, there were no significant changes
in long-range transport patterns. The most likely factors for the observed decreasing trends are declining
anthropogenic emissions due to emission mitigation policies of the European Union.
**1 Introduction**
Epidemiological studies show that increased particulate air pollution due to anthropogenic emissions leads to
adverse effects upon health, including not only respiratory but also cardio-vascular disease (Seaton et al., 1995),
further increases global disease burden (Cohen et al., 2017). Among the ambient sub-micrometer aerosol
(diameter < 1 µm), ultrafine particles (UFP, diameter < 100 nm) share the greatest number fraction of particles.
Previous studies suggested that exposure to UFP might lead to an increased probability of health hazards
(Kreyling et al., 2006; Schmid and Stoeger, 2016), although at present, epidemiological evidence for their effects
upon human health remain mixed due to a number of reasons (Ohlwein et al., 2019). A main rationale for UFP-
driven effects upon health is their ability to penetrate deep into lungs and translocate to other organs such as
brain, cause other health problems such as respiratory and cardiovascular diseases (Kreyling et al., 2006; Schmid
and Stoeger, 2016). In urban areas, a significant fraction of UFP mass consists of black carbon (BC), which is
produced due to incomplete combustion of fossil fuel and biomass and then released to the atmosphere (Chen et
al., 2014; Cheng et al., 2013; Pérez et al., 2010). Since BC may operate as a universal carrier of a wide variety of
toxins such as polycyclic aromatic hydrocarbons (PAH) into the human body, exposure to BC shows strong
relevance with cardiopulmonary morbidity and mortality (Janssen et al., 2012).
To reduce the harmful effects caused by air pollution, emission mitigation policies were implemented around
the world. The European Union (EU) is one of the early regions, where emission reduction policies have been
implemented. The main policy instruments on air pollution within the EU include the Ambient Air Quality
Directives and the National Emission Ceilings Directive. EU emission mitigation legislations are directly
formulated based on sources. In Europe, the main anthropogenic sources to primary aerosol particles are fuel
combustions from industrial installations (power generation, industry, etc.), non-road and road transport, and
domestic heating etc. (European Environment Agency, 2017). Member States of EU were required to draw up a
National Programmes to the Commission (http://ec.europa.eu/environment/air/reduction/implementation.htm).
For example in Germany, the Federal Environment Ministry issued the Federal Emission Control Regulations
(German: Bundes-Immissionsschutzverordnung, BImSchV). To reduce the emission from industrial
installations, the BImSchV requires the permit of construction and operation for some industrial installations, in
accordance with the Federal Emission Control Act. The emission limits from large combustion, such as power
plants, are defined as well. For domestic heating, the unsuitable fuels are listed and their emission values are
defined to control the emission. In Europe, traffic emissions have been found to be a dominant contributor to air
pollution in the urban outdoor atmosphere (Kumar et al., 2010; Pey et al., 2009). Another policy, the clean air
plan (German: Luftreinehalteplan) has great practical importance on the operation of vehicles. It set up low
emission zone (LEZ) in Germany to limit the emission of nitrogen oxide and aerosol particle from the traffic
exhaust. Previous short-term studies indicated that a LEZ can reduce the pollutant concentration immediately
after its implementation, as a result of the absence of the most polluting vehicles (Rasch et al., 2013; Qadir et al.,
2013; Jones et al., 2012).



To evaluate the effectiveness of those emission mitigation policies, long-term observations of pollutants are
crucial, especially for those health-related pollutants, such as sub-micrometer particles and BC. There have been
many studies about long-term trends of particle number concentration (PNC) or BC mass concentration since the
1990s. These studies concluded emission mitigation policies may reduce the human exposure to the pollutants,
and were important for the policy makers (Barmpadimos et al., 2011; Masiol et al., 2018; Kutzner et al., 2018;
Putaud et al., 2014; Sabaliauskas et al., 2012; Wang et al., 2012). However, most of these studies were
conducted at roadside or urban background, which are largely dominated by traffic emissions. Only a few studies
focused on long-term trends of the PNC or the BC mass concentrations at the regional background setting (Asmi
et al., 2013; Barmpadimos et al., 2011; Murphy et al., 2011). Murphy et al. (2011) found that the elemental
carbon (EC) mass concentration decreased in several national parks and other remote sites in the US between
1990 and 2004. This result was an indication that emission control policies were effective in reducing the EC
mass concentration in the background air across the US. Asmi et al. (2013) analysed the long-term change of
PNC at the regional background and remote sites in Europe, North America, Antarctica, and Pacific Ocean
islands. The results showed that decreased PNCs could likely be explained by the reduction of anthropogenic
emission. Kutzner et al. (2018) evaluated the long-term trend of BC over Germany including industrial, rural,
traffic and urban background sites. The result confirmed that emission control policy in the last two decades has
most likely contributed to mitigate BC mass concentration in Germany and Europe. However, the long-term
trend studies of PNC and BC mass concentration measured in parallel and covering different environments from
roadside, urban background to regional background and remote areas in the same region have not been done.
This study takes Germany as an example to understand the effectiveness of emission mitigation policies on
the reduction of the regional PNC and BC mass concentration. In this investigation, trend analysis was done for
the sub-micrometer PNC (diameter < 1 µm) and the equivalent black carbon (eBC) mass concentration in
Germany based on a unique dataset of the German Ultrafine Aerosol Network (GUAN). For the period of study
(2009-2018), 16 observational sites have been included, ranging from roadside to high Alpine. The weekly,
diurnal, seasonal trends and the robustness of the trend were evaluated. To determine, if the past emission
mitigation policies are the decisive factor for the long-term trends of the eBC mass concentration and PNC at
different environments, the influences of other potential drivers (i.e. the meteorological condition change and
long-range transport pattern change) are also discussed.
**2 Dataset and method**
**2.1 The German Ultrafine Aerosol Network (GUAN)**
The sites investigated in this study belong to GUAN, which combines federal and state air quality monitoring
stations, as well as atmospheric observatories from research institutes, aiming at a better understanding of sub-
micrometer PNC and BC with respect to human health and climate impact (Birmili et al., 2016). GUAN is a
specialized network in Germany, which provides continuous measurements including sub-micrometer particle
number size distribution (PNSD) and eBC mass concentration, with diverse environments from roadside, urban
background, regional background, low mountain range to high Alpine.
Table 1 lists the basic information of GUAN sites. The locations of GUAN sites are illustrated in Fig. 1. A
summarized description of GUAN sites is given here with more details are available in Birmili et al. (2016).
Among the 17 sites, eight are located in the state of Saxony: Leipzig-Mitte (LMI), Leipzig-Eisenbahnstraße



(LEI), Leipzig-TROPOS (LTR), Leipzig-West (LWE), Melpitz (MEL), Dresden-Nord (DDN), Dresden-Winckelmann-straße (DDW) and Annaberg-Buchholz (ANA). LMI and LEI are two roadside stations in Leipzig. The former one is located at roadside in an open area in the city center, while the latter one is a street canyon station. The traffic volumes at these two sites are 44 000 and 12 000 vehicles per day, respectively. LTR and LWE are urban background sites located in the city of Leipzig with 10 km apart. LTR is an atmospheric research station operated by Leibniz Institute for Tropospheric Research (TROPOS). The station is situated on the roof of the TROPOS institute building. LWE was located in a park on the premises of a hospital, with negligible traffic influence. Station MEL operated by TROPOS since 1992, is located in farmland about 50 km from Leipzig. Previous studies showed that MEL can represent the regional background atmosphere of Central Europe (Spindler et al., 2013). Two stations are located in the city of Dresden: a roadside station DDN with the traffic volume of ~ 36 000 vehicles per day, and an urban background site DDW, which is 1.7 km away from city centre. ANA is an urban background station for Saxon State Office for Environment, Agriculture and Geology (LfULG), located in the city of Annaberg-Buchholz in the Ore mountain area, about 10 km away from German-Czech border (Schladitz et al., 2015).

Three stations are located in the lowlands of the Northern Germany: Bösel (BOS), Neuglobsow (NEU) and Waldhof (WAL). The urban background site BOS is located in the village of Bösel, about 100 km from the North Sea, it is thus partly influenced by maritime air masses. NEU and WAL are located in forests, representing regional background conditions in the Northern Germany lowlands.

Two stations, Langen (LAN) and Mülheim-Styrum (MST), are located in the west of Germany. LAN is an urban background site located in the city of Langen, at the edge of a residential area and a forest. Emission from the Frankfurt's Rhein-Main airport (about 5 km to the southeast) may influence the observations at LAN. MST is situated in the western end of the Ruhr area, the largest urban area in Germany.

Four stations are located in the south part of Germany, including one urban background site Augsburg (AUG), two low mountain range sites, Schauinsland (SCH) and Hohenpeißenberg (HPB), and one high Alpine site Zugspitze (ZSF, Schneefernerhaus). AUG is located on the premises of Augsburg's University of Applied Sciences about 1 km southeast of Augsburg city center. The two low mountain range sites SCH and HPB are surrounded mainly by forests and agricultural pastures. Their elevations are 1205 and 980 m a.s.l., respectively. The high Alpine site ZSF is located at 2670 m a.s.l., 300 m below the summit of the Zugspitze, at the south side of the highest mountain in Germany.

It needs to be noted that the selection of sites in GUAN could not be designed from scratch. As financial resources to perform specialized air pollution measurements are limited, GUAN has incorporated such sites where sub-micrometer particles were already measured by one of the partner institutions, or sites that could be co-established with the aid of other research projects or programs. This explains the incomplete geographic coverage of Germany with GUAN measurement sites.

**2.2 Instrumentation**

The technical details of the PNSD and the eBC mass concentration measurements at GUAN sites are summarized in this section and Table 2. Details of the instrumentation and data processing techniques are provided in Birmili et al. (2016). Depending on individual set-up, the PNSD are measured either by Mobility Particle Size Spectrometers (MPSS, Wiedensohler et al., 2012) or by Dual Mobility Particle Size Spectrometers (D-MPSS). Regenerative Nafion dryers are used to dry the aerosol sample to a relative humidity below 40 %



(Swietlicki et al., 2008). The PNSD is obtained from the raw mobility distributions by an inversion algorithm
(Pfeifer et al., 2014), including the commonly used bipolar charge distribution (Wiedensohler, 1988).
Corrections for diffusional losses in instruments and inlets were made according to Wiedensohler et al. (2012).
Due to the individual settings of MPSS at GUAN sites, the quality of the PNSD was ensured by onsite or
laboratory inter-comparisons conducted by the World Calibration Center for Aerosol Physics (WCCAP,
http://www.wmo-gaw-wcc-aerosol-physics.org/) at TROPOS. The frequency of quality control is between one to
four times per year, as recommended by Wiedensohler et al. (2018).
Mass concentrations of eBC have been measured by Multi-Angle Absorption Photometers (MAAP, Thermo
Scientific, model 5012), except in AUG where an Aethalometer (Type 8100, Thermo Fisher Scientific Inc.) is
used. For MAAP measurement, eBC mass concentration is obtained using a mass absorption cross section of 6.6
$m^2$ $g^{-1}$ for the wavelength of 637 nm (Petzold and Schönlinner, 2004; Müller et al., 2011). No eBC data are
available for LAN and MST.
To condense the information provided by PNSD, we chose three particle size ranges to obtain integrated
PNCs: 10-30 nm, 30-200 nm, and 200-800 nm. The young Aitken mode $N_{[10-30]}$ represents the particles freshly
formed by homogeneous nucleation from either photochemical processes or downstream of traffic exhausts.
Aitken mode particles $N_{[30-200]}$ are either directly emitted from incomplete combustion or grown by
condensational growth. The accumulation mode $N_{[200-800]}$ represents aged particles, which underwent
condensational growth or cloud processing during long-range transport. Since the particles below 20 nm were
not measured all the time from 2009 to 2018 at ZSF and MST, we use $N_{[20-800]}$ to represent total PNC in this
study instead of $N_{[10-800]}$. Data coverage can largely influence the evaluation of long-term trends. Figure 2
illustrates the data coverage of 16 stations in GUAN until the end of 2018, except LWE. LWE is not evaluated in
this study since its observation shows high similarity with LTR (Sun et al., 2019) and it was terminated at the
end of 2016.
**2.3 Trend analysis methods**
Most of the environmental data are not normally distributed. Therefore, non-parametric methods are often used
to detect the long-term trends (Asmi et al., 2013; Barmpadimos et al., 2011; Bigi and Ghermandi, 2014; Collaud
Coen et al., 2007; Collaud Coen et al., 2013; Mejía et al., 2007; Murphy et al., 2011; Sharma et al., 2006).
Detection of long-term, linear trends might be affected by several factors, such as the time span and time
resolution of available data, the magnitude of variability, autocorrelation and periodicity in the time series
(Weatherhead et al., 1998). To analyse the temporary trend of the PNCs and the eBC mass concentrations, two
trend evaluation methods were used in this study.
**2.3.1 Customized Sen-Theil trend estimator**
The customized Sen-Theil trend estimator (customized Sen's estimator, hereafter) is a modified non-parametric
procedure based on the normal Sen's slope estimator, regardless the influence of outlier, missing values and
statistical distribution (Sen, 1968; Theil, 1992; Birmili et al., 2015). This approach estimates the true slope by
fully considering the effect of some periodic variation of atmospheric pollution, such as seasonal, weekly, or
diurnal cycles. It is thus possible to estimate the true slope by this approach for the shorter data set with higher
time resolution, for example 5-year hourly time series. Based on the hourly or daily time series $x(i)$, firstly, rates
of change $m_{i,k}$ on each data pair $[x(i), x(i + k \times 364\ days)]$ is calculated as:





$\quad m_{i,k} = \frac{(x(i+\Delta t)-x(i))}{\Delta t}$ (1)
with $\Delta t = k \times 364 \ days$.
where $k$ is the integer. $\Delta t$ ensures that each data point can be only compared with data points separated by a
multiple of 52 weeks (= 364 $days$), that is, two data points are compared only if they belong to the same hour of
the day, day of the week, and season of the year. For each time series, some 10000 slope $m_{i,k}$ are calculated. The
median of those slopes $m_{i,k}$ is taken as the true slope $m$ over the whole period. Significance and confidence
interval (CI) of the trends are determined at 95 % confidence level from the distribution of $m_{i,k}$.

### 2.3.2 Generalized Least-Square-regression and Auto-Regressive Bootstrap confidence intervals (GLS/ARB)

The second method used to detect the trend is the Generalized Least-Square-regression (GLS) (Mudelsee, 2010;
Asmi, 2013). For a time series of observation $x(i)$, compactly written as $\{t(i), x(i)\}_{i=1}^{n}$, we separate the time
series as:
$\quad x(i) = \beta_1 + \beta_2 t(i) + \Omega(t(i)) + S(i)e(i)$ (2)
where $\beta_1$ and $\beta_2$ are two trend parameters (intercept and slope), $S(i)$ is the variability function scaling the
random noise term $e(i)$, $\Omega(t(i))$ is the seasonal component. In this study, four seasonal components are defined
as:
$\quad \Omega_1 = \beta_3 \sin(\frac{2\pi t}{(1 \ year)}), \Omega_2 = \beta_4 \sin(\frac{4\pi t}{(1 \ year)}),$
$\quad \Omega_3 = \beta_5 \cos(\frac{2\pi t}{(1 \ year)}), \Omega_6 \Omega_4 = \beta_6 \cos(\frac{4\pi t}{(1 \ year)}).$ (3)
$\quad$ The GLS regresses two trend and four seasonal parameters (denoted as $\boldsymbol{\beta}$, thereafter) by minimizing the sum
of squares:
$\quad SSQG(\boldsymbol{\beta}) = (\boldsymbol{x} - \mathbf{T}\boldsymbol{\beta})' \boldsymbol{V}^{-1} (\boldsymbol{x} - \mathbf{T}\boldsymbol{\beta})$ (4)
where,
$\quad \boldsymbol{\beta} = \begin{bmatrix} \beta_1 \\ \vdots \\ \beta_6 \end{bmatrix}$ (parameter vetor),
$\quad \boldsymbol{x} = \begin{bmatrix} x(1) \\ \vdots \\ x(n) \end{bmatrix}$ (data vetor),
$\quad \boldsymbol{T} = \begin{bmatrix} 1 & t(1) & \Omega_1(t(1)) & \cdots & \Omega_4(t(1)) \\ \vdots & \vdots & \vdots & \ddots & \vdots \\ 1 & t(n) & \Omega_1(t(n)) & \cdots & \Omega_4(t(n)) \end{bmatrix}$ (time matrix),
and $\boldsymbol{V}$ is the covariance matrix. The estimated $\boldsymbol{V}$ matrix is:
$\quad \hat{V}\hat{V}(i_1, i_2) = \hat{S}(i_1) \times \hat{S}(i_2) \times \exp[-|t(i_1) - t(i_2)|/\hat{\tau}'], (i_1, i_2 = 1,\dots,n)$ (5)
$\hat{S}(i_1), \hat{S}(i_2)$ are the variability of time series at $t(i_1), t(i_2)$. Here $S$ is assumed to be time invariant, therefore $\hat{S}(i)$
is the stand deviation of the observation time series $x(i)$. $\hat{\tau}'$ is the estimated, bias-corrected persistence time. To
estimate the persistence time, the least-squares estimation is defined:
$\quad S(\tilde{\tau}) = \sum_{i=1}^{n} [x_{\text{noise}}(i) - \exp\{-[t(i) - t(i-1)]/\tilde{\tau}\} \times x_{\text{noise}}(i-1)]^2$ (6)
and $\hat{\tau} = \text{argmin}[S(\tilde{\tau})]$. The minimization of $S(\tilde{\tau})$ is done by Brent's search (Press et al., 1992).
$\quad$ After obtaining the covariance matrix $\boldsymbol{V}$, the solution of Eq.(6) is the GLS estimator:
$\quad \hat{\boldsymbol{\beta}} = (\boldsymbol{T}'\boldsymbol{V}^{-1}\boldsymbol{T})^{-1} \boldsymbol{T}'\boldsymbol{V}^{-1}\boldsymbol{x}$ (7)



Firstly, initial estimation of parameters $\boldsymbol{\beta}$ are approximated. According to the estimated $\boldsymbol{\beta}$, the trend, seasonal
and noise component are obtained from $x(i)$. Then, the persistence time $\hat{\tau}'$ and covariance matrix $\boldsymbol{V}$ are updated
to iterate the GLS fitting until the relative difference between the $\boldsymbol{\beta}$ from last two iterations is below a threshold

236    0.01 %.

To evaluate the robustness of the estimated slopes, the Auto-Regressive Bootstrap (ARB) was used to
construct the confidence intervals (CIs) of the slopes (Mudelsee, 2010, algorithm 3.5). Firstly, the residual $e(i)$
and persistence time $\tilde{\tau}$ are calculated from GLS approach. Then, ARB resamples the white-noise residuals of
data by using the auto-regressive persistence model AR(1), adds the resampled residuals to fitted data and re-
calculates the slopes. The resampling was repeated 1000 times and the CIs were estimated from these 1000
resampled slopes.
To ensure the comparability of trend slopes among different sites, the relative slope in % per year from both
methods is used by dividing the absolute slope by the fitted median value of the first year.

### 2.3.3 Regional Mann-Kendall test

The Mann-Kendall test is a commonly used method to detect the long-term trend (Mann 1945; Kendall 1938). It
detects the trend by Kendall's tau test, which is known as a rank correlation test and it evaluates if a monotonic
increasing or decreasing trend exists. If a significant monotonic increase or decrease is detected, a Sen's slope
estimator is further used to determine the slope and CI of the corresponding time series based on Mann-Kendall
test (Gilbert, 1987). To detect if an overall increase or decrease exists in a multi-site dataset, the regional Mann-
Kendall test was extended to detect the trend over an observation network (Helsel and Frans, 2006).
For a time series $x(i)$ of length $n$, the ordinary Mann-Kendall statistic $S$ is defined as
$$S = \sum_{k=1}^{n-1}\sum_{j=k+1}^{n} \text{sgn}(x(j) - x(k)) \tag{8}$$
where
$$\text{sgn}(\theta) = \begin{cases} 1 & \text{if } \theta > 0 \\ 0 & \text{if } \theta = 0 \\ -1 & \text{if } \theta < 0 \end{cases} \tag{9}$$
For large sample size ($n>10$), $S$ is converted to a normal test statistic $Z$:
$$Z = \begin{cases} \frac{S-1}{\delta_S} & \text{if } S > 0 \\ 0 & \text{if } S = 0 \\ \frac{S+1}{\delta_S} & \text{if } S < 0 \end{cases} \tag{10}$$
where the standard deviation of $S$ is:
$$\delta_S = \sqrt{(n/18)(n-1)(2n+5)} \tag{11}$$
A positive or negative Z refers to a monotonic increasing or decreasing trend. The significance of the trend
can be evaluated by a two-tail test. At α = 0.05 significance level, the null hypothesis of no trend is rejected if
$|Z|>1.96$.
Taking account of multi-sites, the regional Mann-Kendall test evaluates the individual Mann-Kendell statistic
$S_k$ on each individual site $k$ separately by Eq.(8), and sums of individual $S_k$ to obtain a regional Mann-Kendall
statistic $S_L$ and then, $Z_L$ can be obtained:
$$Z_L = \begin{cases} \frac{S_L-1}{\delta_L} & \text{if } S_L > 0 \\ 0 & \text{if } S_L = 0 \\ \frac{S_L+1}{\delta_L} & \text{if } S_L < 0 \end{cases} \tag{12}$$





where the standard deviation of $S_L$ is:
$\delta_L = \sqrt{\sum_{k=1}^{m}(n_k/18)(n_k-1)(2n_k+5)}$          (13)
and $n_k$ is the number of the data at $k$th site.
Once the significant trend is detected, the slope can be evaluated by ordinary Sen's slope estimator (Sen,
1968). For a time series $x_i$, the Sen's slope $m_L$ at each site $L$ is:
$m_L = \frac{1}{n}\sum_{k=1}^{n-1}\sum_{j=k+1}^{n}\frac{x(j)-x(k)}{j-k}$          (14)
Then, the overall Sen's slope $m$ is obtained by the median of those $m_L$. Considering the dataset size and
calculation efficiency, the monthly median time series was used for the regional Mann-Kendall test in this study.
**3 Trends results over the whole time period 2009-2018**
**3.1 Overall trends**
The temporal trends of the PNCs and eBC mass concentrations were evaluated by the customized Sen's
estimator and GLS/ARB. For the customized Sen's estimator, the daily median time series were used, while the
monthly median time series for GLS/ARB. The relative annual slopes are shown in Table 3. Firstly, for 5
parameters at 16 sites (77 trends in total), two trend detection methods agree with each other very well, with six
exceptions: $N_{[20-800]}$ at MST, $N_{[10-30]}$ at BOS, HPB and SCH, and $N_{[30-200]}$ at LAN and HPB, which we conclude as
no increase or decrease. In general, significant decrease of the eBC mass concentration and $N_{[200-800]}$ are detected
at all evaluated sites, except LAN, where no significant trends were found. The slopes of $N_{[10-30]}$ show high
variability and lowest number of significant trends: at 7 sites there is a significant decrease and only MEL
increase for both trend methods. Significant decrease of $N_{[30-200]}$ was found at all sites except LAN and three
other regional background and mountain sites (MEL, NEU, HPB). In general, the annual slope of the eBC mass
concentration varies between -7.7 % and -1.8 % per year, and the slope of the PNCs varies from -6.3 % to 2.7 %,
-7.0 % to -2.0 %, and -9.5 % to -1.5 % per year (only significant trends) for 10-30 nm, 30-200 nm, and 200-800
nm, respectively. At site LAN, only insignificant decreases of the PNCs were detected. One speculation is that,
due to its low data coverage at LAN, the trend detection methods might be hard to find the significant change.
To detect if there is decrease of eBC mass concentration and PNC at LAN, we evaluated the trend of eBC mass
concentration at another urban background site Raunheim. Site Raunheim is an urban background site of
German Environment Agency (UBA), located in the city of Raunheim, about 15 km far away from LAN. The
slopes of eBC mass concentration at Raunheim are -7.2 % and -5.9 % per year (both significant) for the
customized Sen's estimator and GLS/ARB, respectively. It could be an indicator for reduction of eBC mass
concentration at LAN.
On one hand, for diverse pollutant parameters and sites, their spatial representativeness is different due to the
lifetime of pollutant and local influence (Sun et al., 2019). On the other hand, as shown in Table 3, not all the
sites show the significant decreases of PNCs. Therefore, it is hard to conclude the regional reduction of the eBC
mass concentration and PNCs all over Germany from the slopes evaluated at individual sites. To evaluate the
regional variation of the eBC mass concentration and PNCs all over Germany, the regional Mann-Kendall trend
is shown in Table 3 as well. It should be noted that, three roadside sites might bias the result of regional Mann-
Kendall test due to their prominent local influence. Moreover, the locations of the other 13 sites in GUAN are
not evenly distributed in spatial scale since there are 5 sites located in the state of Saxony and HPB and ZSF are





only 42 km apart from each other. This will result in a false trend throughout the entire region. To ensure the
representativeness of spatial sampling, three roadside sites (DDN, LMI and LEI) as well LTR, ANA and ZSF
are excluded in the regional Mann-Kendall test.

The highest regional reduction rate appears on the eBC mass concentration of which anthropogenic

emissions are the major source in Germany. The regional trends of the PNCs in the size ranges 30-200 nm and
200-800 nm are both significantly negative. $N_{[30\text{-}200]}$ represents the particles originated from anthropogenic
emissions and the aged particles from new particle formation (NPF). Especially at urban area, $N_{[30\text{-}200]}$ and eBC
mass concentration are found to be closely related to the emissions from incomplete diesel combustion (Cheng et
al., 2013; Krecl et al., 2015). Significant regional decrease of $N_{[30\text{-}200]}$ and eBC mass concentration might indicate
that, declined anthropogenic emission is an important or even dominant driver for those decreases in Germany.
Insignificant regional trend was detected for 10-30 nm. One explanation could be anthropogenic emissions have
probably only minor or negligible influence on $N_{[10\text{-}30]}$ at the regional background area due to the short lifetime
and high spatial variability of young Aitken mode particles (Sun et al., 2019).

The trends of the PNC and eBC mass concentrations in this study are in consistent with studies in other

European countries. In Europe, the negative trends of the total PNC, particle light absorption coefficient, and
other optical properties were found at 9 regional background or remote sites from 2000 to 2010 (Asmi et al.,
2013; Collaud Coen et al., 2013). In Spain, the $PM_{10}$ and $PM_{2.5}$ decreased about -5.9 % and -6.0 % from 2004 to
2014, respectively (Pandolfi et al., 2016). The significant decrease of $PM_{10}$ has been detected since 2008 due to
the influence of reduced primary anthropogenic emissions in Po Valley, one large industrial manufacturing
district in Europe (Bigi and Ghermandi, 2016). A similar study was conducted in UK. The BC trend from 2009
to 2016 varied between -0.62 % and -8 % at street, urban and rural background sites (Singh et al., 2018).
**3.2 Robustness of the trends**
For the time series of a climate parameter, its trend may be caused by the homogenous variations in
meteorological conditions or aerosol emissions (Conrad and Pollak, 1950), but sometimes also can be caused by
inhomogeneous "break points" such as site relocation, inlet change, and new pollution sources (Collaud Coen et
al., 2013). The break points not only make the time series inhomogeneous but also result in a poor
representativeness of the trend. Normally, only the trends of homogenous time series are considered to be robust
and trustable. Another important factor affecting the trend is the size of the time series. As shown in Fig. 2, the
sizes of the time series are not the same for all evaluated sites, vary from 6 to 10 years. To evaluate if the
detected decreases or increases are homogeneous and if our dataset is long enough to provide the robust trend,
the evolution of trend was analyzed. Fig. 3 shows the annual changes of the eBC mass concentration and PNCs
for expanding time intervals starting from 2009, using the customized Sen's estimator. The average trend for
each site category is illustrated. It can be seen that, the trends tend to be stable without strong variation after time
interval 2009-2016, indicating our dataset is sufficient for true slopes.

Gaps in time series may bias the observed trends. Generally, it is difficult to quantify clearly the influence of

data gaps on the trend results. In this study, since the influences of periodicity and outliers are diminished by the
customized Sen's estimator, the evaluated trends are less sensitive to data gaps than those derived by other
methods. Still, data gaps may affect the trend results especially for sub-dataset, for example the trends in
particular seasons.





## 4 Trend in sub-sets

As shown in Sect. 3, declined anthropogenic emissions are very likely to be the main factor of the decreased PNCs and eBC mass concentration in Germany. The intensity of human activities such as traffic volume usually has weekly and diurnal cycles. To further investigate the role of anthropogenic emissions in the downward trend of the PNCs and eBC mass concentration, their weekly, diurnal and seasonal trends were analyzed in this section.

### 4.1 Weekly trends

For the weekly Sen's slope, only the data pairs belonging to the same weekday were selected to calculate the slope $m$. Figure 4 illustrates the average Sen's slopes of the PNCs and eBC mass concentration for working day (from Monday to Friday) and weekend (Saturday and Sunday) at each site category.

At roadside where traffic emission dominates the PNCs and eBC mass concentration, higher reduction rates are observed on working days for all five parameters. Traffic emission has direct influence on urban background aerosol, thus reduction rates at urban background sites are higher on weekday. But the differences are smaller than those for roadside. This result implies that traffic emission control policies such as LEZ is a main factor of the decreases of the PNCs and eBC mass concentration in urban area. There is no significant difference can be seen between working day and weekend for the regional background, low mountain range and high Alpine sites, rather indicating that the cause for the decrease is far away from the background condition and hence closer to urban areas.

### 4.2 Diurnal trends

Figure 5 shows the customized Sen's slopes of the PNCs and eBC mass concentration at each hour of day. Similar to the weekly trend, data pairs belonging to a particular hour of day were selected to calculate the slope $m$.

For BC which is mainly emitted from anthropogenic sources in Europe, diurnal patterns with higher reduction rate in daytime than in night-time can be seen at roadside sites. Reduction of traffic emission can directly cause a decrease of eBC mass concentration in near source areas. Therefore, higher reduction rate is observed in daytime when human activities are more intensive. Negative slopes can be also observed in night time and in other site categories. A plausible explanation is that, reduction of local anthropogenic emissions can also reduce the background eBC mass concentration in a larger area and longer time scale since BC has a lifetime of around a week (Cape et al., 2012; Wang et al., 2014). This result confirms that reduction of anthropogenic emissions plays a main role in the decreasing trends of eBC mass concentration in Germany.

The trends of the PNCs depend on the particle size ranges and time of day. In most of roadside sites, similar diurnal patters as that for eBC with higher reduction rate in daytime and lower rate in nighttime can be observed for $N_{[20-800]}$, $N_{[10-30]}$ and $N_{[30-200]}$. In cities, traffic emission may have large contribution on PNC in these size ranges, thus we attribute this diurnal pattern of reduction rate also to the reduced traffic emission in urban background conditions, similar as to eBC mass. NPF is an important natural source of ultrafine particles and may largely enhance $N_{[10-30]}$. Based on the GUAN dataset, Ma and Birmili (2015) reported that the annual average contributions of NPF on $N_{[5-20]}$ are 12 %, 24 % and 54 % at roadside, urban background and regional background sites, respectively. Therefore, the inter-annual change of NPF frequency or intensity may also determine the trend of $N_{[10-30]}$ especially in urban and regional background sites. Actually, as can be seen in Fig. 5c that $N_{[10-30]}$ show a maximum reduction rate of around -3 % in the afternoon at the regional background sites. It is likely to





be resulted from the inter-annual change of regional NPF events since NPF is the only dominant source at those
sites. At regional and mountain sites, $N_{[30-200]}$ and $N_{[200-800]}$ show a constant negative trend throughout the day,
suggesting the decrease of PNCs in the regional background air which is likely to be the result of the reduction
of anthropogenic emissions in cities.

**4.3 Seasonal trends**

It is obvious that the seasonal change of weather condition will have an influence on the change of PNCs and
eBC mass concentration. In the warm season, the higher plenary boundary layer (PBL) height and better dilution
can reduce the PNCs and eBC mass concentration, but NPF events may increase the PNC especially the
nucleation mode particles $N_{[10-30]}$. Conversely, PNC and eBC mass concentrations are elevated in cold season
due to a less mixed PBL and higher anthropogenic emissions such as domestic heating. In this section, the
seasonal trends of the eBC mass concentration and PNCs were detected. For seasonal trends, only the data pairs
belonging to a particular season were used to calculate the customized Sen's slope $m$.
Figure 6 shows the statistical results of the multi-annual trends of the PNC and eBC mass concentrations for
different seasons. In general, negative trends are found in all sites and pollutant parameters except $N_{[10-30]}$.
Reductions of the PNC are found to be stronger in winter, which can be regarded as a result of the
implementation of the emission mitigation regulations for large or small combustions, such as domestic heating
or power generation in winter. Conversely, the least decreases of the PNCs were found in summer. One impact
factor might be the seasonal variation of biogenic emission (Asmi et al., 2013). The biogenic emission increases
in summer, which will mask the decrease caused by anthropogenic emission. In winter, less biogenic emission
makes anthropogenic emissions more prominent. Therefore, a higher decrease can be seen in winter, indicating
that the decreasing trends of the PNCs are more likely related to anthropogenic sources than biogenic ones.
Long-term change of meteorological parameters may affect the seasonal trend as well. It will be discussed in the
next section.

**5 Meteorological influence on the trend of the particle number and the eBC mass concentration**

Meteorological conditions also influence the temporal variation of aerosol particles (Birmili et al., 2001;
Mikkonen et al., 2011; Spindler et al., 2013; von Bismarck-Osten et al., 2013; Wehner and Wiedensohler, 2003;
Hussein et al., 2006). Long-term changes of meteorological conditions (precipitation, PBL height, wind speed,
temperature etc.) could cause increase or decrease of atmospheric pollutant concentration. To investigate the
contribution of possible changes in meteorological conditions in the period of interest, trends under different
weather conditions are discussed in this section.

**5.1 Seasonal trends of meteorological parameters**

Table 4 provides the long-term trends of precipitation, ambient temperature, and wind speed all over Germany
for the period 2009-2018. The meteorological data was obtained from Germany's National Meteorological
Service (Deutscher Wetterdienst, DWD). The daily values of these three meteorological parameters at 76
measuring sites in Germany were provided. The mean time series among all 76 sites were used as the area
average of meteorological data in Germany. The trends of meteorological parameters were evaluated by the
customized Sen's estimator. Firstly, the significant slope of precipitation was found only in summer, -5.9 % per





year. Decreased precipitation in summer might result in less wet deposition and thus in a smaller reduction rate
of the eBC mass concentration and $N_{[200-800]}$. For the ambient temperature, the significant increases were detected
in summer, autumn and winter. Increased temperature, especially in winter, may lead to lower anthropogenic
emissions from domestic heating or power generation. In addition, slight increase of wind speed was observed in
winter, resulting in an increased dilution and thus decreased pollutant concentrations. In summary, increased
ambient temperature and wind speed in winter might contribute to the decrease of the PNCs and eBC mass
concentrations. However, decreased precipitation in summer may diminish the decrease of the PNCs and eBC
mass concentration.
**5.2 Air mass dependency on long-term changes in particle number concentration and equivalent black**
**carbon**
Synoptic-scale air masses, representing different weather conditions and long-range transport pattern, can be
used to explain the different temporal variation of aged aerosol particles (Ma et al, 2014; Hussein et al., 2006).
Two factors may control the concentration of aerosol particles in different air masses: residence time over the
continent and regional emission at origin region.
To investigate the influence of the long-range transport pattern, a backward trajectory clustering method was
used. This method, denoted as back-trajectory cluster method (BCLM), is based on a joint cluster analysis
considering backward trajectories, $PM_{10}$ mass concentration, and profiles of pseudo-potential temperature at
several sites over Germany, including regional background, low mountain range and high Alpine conditions
(Birmili et al., 2010; Engler et al., 2007; Ma et al., 2014). In this study, 15 air mass types are obtained from
BCLM to represent the overall meteorological condition on a large scale over Germany, and it is thus valid for
all GUAN sites. It should be noticed that, the time span of BCLM is from 2009 to 2014, which does not totally
cover the whole observation time in our trend analysis (2009-2018). However, as the trend evolution plots in Fig.
3 and one previous short-term study (2009-2013) of GUAN dataset (Birmili et al., 2015) shown, reductions of
the PNCs and eBC mass concentration have been observed at most of GUAN sites during 2009-2014. Therefore,
to evaluate the influence of long-range transport on the decrease of the PNCs and eBC mass concentration, the
BCLM was used in this section since we believe the change of long-range transport pattern in 2009-2014 could
represent its change in the whole time period (2009-2018). More information about data preparation, cluster
processing, and data procedures and data products is described in detail in a corresponding research article by
Ma et al. (2014). Figure 7 shows the average trajectories and the average normalized profiles of pseudo potential
temperature ($\theta_v$) for each air mass type. According to their vertical stability and meteorological condition as
shown in Fig. 7b, the 15 air mass types are named by the season and atmospheric flow: CS: cold season; TS:
transition season; WS: warm season; ST: Stagnant; A: Anti-cyclonic; C: cyclonic. The vertical stability is more
stable at CS air masses and more neutral in WS air masses. Table 5 lists the basic statistical information of each
air mass type.
**5.2.1 Particle number concentration and equivalent black carbon mass concentration for each air mass**
**type**
Figure 8 illustrates the median value of the PNCs and eBC mass concentrations with respect to the 15 air mass
types at regional background site category (MEL, WAL and NEU). First, there is less significant difference on
$N_{[10-30]}$ and $N_{[30-200]}$ among different air mass types, since $N_{[10-30]}$ and $N_{[30-200]}$ represent more local information.
For the $N_{[200-800]}$ and eBC mass concentration, higher values are observed in CS air masses as shown in Fig. 8a





and 8e. This can be explained by higher anthropogenic emissions and less dilution caused by lower PBL height
in cold season. In the same season (WS, CS or TS), the median values of the $N_{[200-800]}$ and eBC mass
concentration differ with regard to atmospheric air flows. The $N_{[200-800]}$ and eBC mass concentration at the air
mass types A1 and ST (CS-A1, WS-A1, TS-A1, CS-ST, and WS-ST), are always higher than the ones at other
air mass types in the same season. Because these air masses remained as least three days over Central Europe
before reaching the measurement sites. During these three days, emitted aerosol particles are continuously
accumulated into the air masses. Within these five air mass types, the median values of the $N_{[200-800]}$ and eBC
mass concentration at CS-A1 and WS-A1 are relatively higher, since the anti-cyclonic air mass usually comes
from Eastern Europe with more anthropogenic emissions. Moreover, the median values of the $N_{[200-800]}$ and the
eBC mass concentration at the air masses type A2 (CS-A2, TS-A2, and WS-A2), C1 and C2 decrease steadily,
which can be explained by the shorter residence time over the European continent.
**5.2.2 Influence of air masses frequency change on the trend of the particle number concentration and the**
**equivalent black carbon mass concentration**
As shown in Fig. 8, the PNCs and eBC mass concentration vary widely with respect to different air mass types,
meaning the air mass type is one of the factors to change the pollutant concentrations, especially for aged
accumulation mode particles $N_{[200-800]}$ and eBC mass concentration. Therefore, the frequency change of air
masses might lead to a change of long-term trend of the PNCs and eBC mass concentration. In this section, the
relationship between air mass frequency change and concentration change is discussed.
It is, however, hard to detect the long-term trend of pollutant parameters for each individual air mass type
because the frequency of air masses varies in a range of 2.6 % to 12.4 % (see Table 5). This means that some of
them are too sensitive to detect the temporal change since their frequencies are too low. Therefore, it is needed to
further group the 15 air mass types. According to the different eBC mass concentration values at different air
mass types (see Fig. 8a), the 15 different air mass types are grouped into two categories:
(1): Polluted air mass category includes CS-ST, CS-A1, CS-A2, CS-C1, TS-A1, WS-ST, WS-A1, and WS-C1;
(2): Cleaner air mass category includes CS-C2a, CS-C2b, TS-A2, TS-C1, TS-C2, WS-A2, and WS-C2.
Figure 9 shows the relationship between air mass frequency change and mean pollutant concentration change
at all regional background and low mountain range sites, with respect to each air mass category. If the air mass
frequency change is an dominate factor for the downward trend of BC and accumulation mode particle $N_{[200-800]}$,
a decrease in polluted air mass frequency should be associated with a decrease in $N_{[200-800]}$ and eBC mass
concentration. From Fig. 9a, the frequency of polluted air mass does not consistently decrease: It slightly
decreased from 2009 to 2012, and then started to increase after 2012. However, the annual mean values of the
PNCs and the eBC mass concentrations consistently decrease at both air mass categories for all parameters.
Therefore, it can be concluded that the change of long-range transport pattern is not the reason causing the
reduction of pollutant concentrations.
To sum up, the long-term change of meteorological parameters and long-range transport pattern are analyzed
in this section to investigate their contribution to the downward trend of the PNCs and eBC mass concentration
in Germany. The results show that increased ambient temperature and wind speed in winter since 2009 are
thought to have a contribution to declined eBC mass concentration and PNCs, as a result of less anthropogenic
emissions from domestic heating etc. and slightly stronger dilution by higher wind speed. However, decreased
precipitation in summer may diminish the decrease of the PNCs and eBC mass concentration. Moreover, the





change of air mass frequency was detected and the results indicate that the change of long-range transport pattern
is not the factor causing the reduction of pollutant concentrations. It is an indication that, the stringent emission
mitigation policies in Germany and Europe have a beneficial effect on the declined eBC mass concentrations and
PNCs.
**6 Conclusion**
In this work, long-term trends of atmospheric particle number concentrations (PNC) and the equivalent black
carbon (eBC) mass concentration over a 10-year period (2009-2018) were determined for 16 sites in the German
Ultrafine Aerosol Network (GUAN), ranging from roadside to high Alpine environment. Overall, significant
downward trends were found for most of these parameters and observation sites. Concretely, the annual slopes of
the eBC mass concentration of all 16 sites varies between -7.7 % and -1.8 % per year, and the significant slopes
of the PNCs vary from -6.3 % to 2.7 %, -7.0 % to -2.0 %, and -9.5 % to -1.5 % per year for particles with
diameters of 10-30 nm, 30-200 nm, and 200-800 nm, respectively. The regional Mann-Kendall test yielded
regional-scale trends of eBC mass concentration, $N_{[30-200]}$ and $N_{[200-800]}$ of -3.8 %, -2.0 % and -2.4 %, respectively,
indicating an overall decreasing trend for sub-micrometer PNC (except $N_{[10-30]}$) and eBC mass concentration all
over Germany. Particularly, the highest regional decrease appears for the eBC mass concentration for which
combustion processes from motor traffic and power generation are the major source in Germany. This implies
that decreasing anthropogenic emissions might be one of the factors causing the reduction of the PNCs and eBC
mass concentrations.
The highest decrease of eBC mass concentration was observed during both working days (from Monday to
Friday) and daytime (06:00-18:00 LT) at roadside and urban background, which implies a strong evidence of
reduced traffic emissions in urban area. As traffic volumes near those sites have changed little in comparison,
our results are indicative of reductions in specific emission factors, facilitated e.g. by the introduction of diesel
particle filters. At regional and mountain sites, most of the trends showed a constant decrease during the whole
week and entire day, rather indicating that the sources for the decrease are far away from the regional
background or mountains and closer to urban areas.
Meteorological conditions are also able to influence the temporal variation of aerosol particles. Seasonal
trends show that the reduction of the PNCs and eBC mass concentrations occurs all year round, however,
stronger in wintertime. There are three explanations for this result:
a) The influence of reduced anthropogenic emission on PNC is thought to be much more prominent in winter

than in summer (Asmi et al., 2013),

b) Increased ambient temperature and wind speed in winter are also thought to have a contribution on

declined eBC mass concentration and PNCs, as a result of less anthropogenic emissions from domestic

heating etc. and stronger dilution,

c) Decreased precipitation in summer might result in less wet deposition and thus less scavenging and a

smaller reduction rate of eBC mass concentration and $N_{[200-800]}$.

Moreover, the change of air mass frequency was determined but the results indicate that the change of long-
range transport pattern is not associated with the reduction of pollutant concentrations. We therefore conclude
that the declining anthropogenic emissions are the most likely decisive factor for the decrease of the eBC mass
concentration and PNCs all over Germany.





This study suggests that a combination of emission mitigation policies can effectively improve the air quality on large spatial scales such as in Germany. Given the relative novelty of the long-term measurements (particle number size distributions, BC) in a network such as GUAN, the results proved to be robust and comprehensive. Our study shows that long-term measurements of aerosol parameters in different environments can be instrumental in detecting and understanding the long-term effects of emission mitigation policies.

**Acknowledgement.** We acknowledge funding by the German Federal Environment Ministry (BMU) grants F&E 370343200 (German title: Erfassung der Zahl feiner und ultrafeiner Partikel in der Außenluft) from 2008 to 2010, and F&E 371143232 (German title: Trendanalysen gesundheitsgefährdender Fein- und Ultrafeinstaubfraktionen unter Nutzung der im German Ultrafine Aerosol Network (GUAN) ermittelten Immissionsdaten durch Fortführung und Interpretation der Messreihen) from 2012 to 2014. For the MST (Mülheim-Styrum) measurements, we thank the co-funding by the North Rhine-Westphalia Agency for Nature, Environment and Consumer Protection (LANUV). Measurements at Annaberg-Buchholz were supported by the EU-Ziel3 project UltraSchwarz (German title: Ultrafeinstaub und Gesundheit im Erzgebirgskreis und Region Usti), grant 100083657. Measurements at DDW (Dresden-Winckelmannstraße) were co-funded by the European Regional Development Fund Financing Programme Central Europe, grant No. 3CE288P (UFIREG). Measurements in AUG (Augsburg) were funded partly also by UFIREG and by the Helmholtz-Zentrum.

The authors would like to thank the technical and scientific staff members of the stations included in this study. André Sonntag and Stephan Nordmann (TROPOS/UBA) contributed to data processing. Prof. Dr. Thomas A.J. Kuhlbusch and Dr. Ulrich Quass contributed the data quality assurance and data analysis at MST. Horst-Günther Kath (State Dept. for Environmental and Agricultural Operations in Saxony, Betriebsgesellschaft für Umwelt und Landwirtschaft – BfUL), Andreas Hainsch (Labour Inspectorate of Lower Saxony, Staatliches Gewerbeaufsichtsamt Hildesheim – GAA), and Dieter Gladtke (Agency for Nature Protection, the Environment, and Customer Protection in North Rhine-Westfalia, Landesamt für Natur, Umwelt und Verbraucherschutz Nordrhein-Westfalen – LANUV) made the GUAN measurements possible at their respective observations sites. We also thank Werner Wunderlich in Hessian State Office for Nature Conservation, Environment and Geology for the eBC mass concentration data at Raunheim and Karin Uhse at German Environment Agency (UBA) for the PNCs data quality check at LAN. We thank Andreas Rudolph, Dustin Konzack and Andreas Fischer at TOPAS GmbH, Dresden, for kindly providing the UFP-monitor TSI 3031 at LAN (data 2015 – 2018), and yearly quality assurance checks.

This work was also accomplished in the frame of the project ACTRIS-2 (Aerosols, Clouds, and Trace gases Research InfraStructure) under the European Union—Research Infrastructure Action in the frame of the H2020 program for "Integrating and opening existing national and regional research infrastructures of European interest" under Grant Agreement N654109 (Horizon 2020). Additionally, we acknowledge the WCCAP (World Calibration Centre for Aerosol Physics) as part of the WMO-GAW program base-funded by the UBA.

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

Theil, H.: A Rank-Invariant Method of Linear and Polynomial Regression Analysis, in: Henri Theil's
Contributions to Economics and Econometrics: Econometric Theory and Methodology, edited by: Raj, B.,
and Koerts, J., Springer Netherlands, Dordrecht, 345-381, 1992.
von Bismarck-Osten, C., Birmili, W., Ketzel, M., Massling, A., Petäjä, T., and Weber, S.: Characterization of
parameters influencing the spatio-temporal variability of urban particle number size distributions in four
European cities, Atmos. Environ., 77, 415-429, http://dx.doi.org/10.1016/j.atmosenv.2013.05.029, 2013.
Wang, Q., Jacob, D. J., Spackman, J. R., Perring, A. E., Schwarz, J. P., Moteki, N., Marais, E. A., Ge, C., Wang,
J., and Barrett, S. R.: Global budget and radiative forcing of black carbon aerosol: Constraints from pole-to-
pole (HIPPO) observations across the Pacific, Journal of Geophysical Research: Atmospheres, 119, 195-206,

751     2014.

Wang, Y., Hopke, P. K., Rattigan, O. V., Chalupa, D. C., and Utell, M. J.: Multiple-year black carbon
measurements and source apportionment using Delta-C in Rochester, New York, J. Air Waste Manage.
Assoc., 62, 880-887, 10.1080/10962247.2012.671792, 2012.
Weatherhead, E. C., Reinsel, G. C., Tiao, G. C., Meng, X. L., Choi, D., Cheang, W. K., Keller, T., DeLuisi, J.,
Wuebbles, D. J., and Kerr, J. B.: Factors affecting the detection of trends: Statistical considerations and
applications to environmental data, Journal of Geophysical Research: Atmospheres (1984–2012), 103,

758     17149-17161, 1998.

Wehner, B., and Wiedensohler, A.: Long term measurements of submicrometer urban aerosols: statistical
analysis for correlations with meteorological conditions and trace gases, Atmos. Chem. Phys., 3, 867-879,
10.5194/acp-3-867-2003, 2003.
Wiedensohler, A.: An approximation of the bipolar charge distribution for particles in the submicron range, J.
Aerosol Sci., 19, 387-389, 1988.
Wiedensohler, A., Birmili, W., Nowak, A., Sonntag, A., Weinhold, K., Merkel, M., Wehner, B., Tuch, T.,
Pfeifer, S., Fiebig, M., Fjäraa, A. M., Asmi, E., Sellegri, K., Depuy, R., Venzac, H., Villani, P., Laj, P.,
Aalto, P., Ogren, J. A., Swietlicki, E., Williams, P., Roldin, P., Quincey, P., Hüglin, C., Fierz-Schmidhauser,
R., Gysel, M., Weingartner, E., Riccobono, F., Santos, S., Grüning, C., Faloon, K., Beddows, D., Harrison,
R., Monahan, C., Jennings, S. G., O'Dowd, C. D., Marinoni, A., Horn, H. G., Keck, L., Jiang, J., Scheckman,
J., McMurry, P. H., Deng, Z., Zhao, C. S., Moerman, M., Henzing, B., de Leeuw, G., Löschau, G., and
Bastian, S.: Mobility particle size spectrometers: harmonization of technical standards and data structure to
facilitate high quality long-term observations of atmospheric particle number size distributions, Atmos.
Meas. Tech., 5, 657-685, 10.5194/amt-5-657-2012, 2012.
Wiedensohler, A., Wiesner, A., Weinhold, K., Birmili, W., Hermann, M., Merkel, M., Müller, T., Pfeifer, S.,
Schmidt, A., and Tuch, T.: Mobility particle size spectrometers: Calibration procedures and measurement
uncertainties, Aerosol Science and Technology, 52, 146-164, 2018.



**Table 1: Basic information of the atmospheric measurement sites in German Ultrafine Aerosol Network (GUAN), in alphabetic**
**order (Birmili et al., 2016).**

| No. | Site name | Abbreviation | Status (Until 2017) | Site category | Elevation | Location |
|---|---|---|---|---|---|---|
| 1 | Annaberg-Buchholz | ANA | In operation | Urban background | 545 m | 50°34'18" N, 12°59'56" E |
| 2 | Augsburg | AUG | In operation | Urban background | 485 m | 48°21'29" N, 10°54'25" E |
| 3 | Bösel | BOS | Terminated end of 2014 | Urban background | 17 m | 52°59'53" N, 07°56'34" E |
| 4 | Dresden-Nord | DDN | In operation | Roadside | 116 m | 51°03'54" N, 13°44'29" E |
| 5 | Dresden-Winckelmann-straße | DDW | In operation | Urban background | 120 m | 51°02'10" N, 13°43'50" E |
| 6 | Hohenpeißenberg | HPB | In operation | Low mountain range | 980 m | 47°48'06" N, 11°00'34" E |
| 7 | Langen | LAN | In operation | Urban background | 130 m | 50°00'18" N, 08°39'05" E |
| 8 | Leipzig-Eisenbahnstraße | LEI | In operation | Roadside | 120 m | 51°20'45" N, 12°24'23" E |
| 9 | Leipzig-Mitte | LMI | In operation | Roadside | 111 m | 51°20'39" N, 12°22'38" E |
| 10 | Leipzig-TROPOS | LTR | In operation | Urban background | 126 m | 51°21'10" N, 12°26'03" E |
| 11 | Leipzig-West | LWE | Terminated end of 2016 | Urban background | 122 m | 51°19'05" N, 12°17'51" E |
| 12 | Melpitz | MEL | In operation | Regional background | 86 m | 51°31'32" N, 12°55'40" E |
| 13 | Mülheim-Styrum | MST | In operation | Urban background | 37 m | 51°27'17" N, 06°51'56" E |
| 14 | Neuglobsow | NEU | In operation | Regional background | 70 m | 53°08'28" N, 13°01'52" E |
| 15 | Schauinsland | SCH | In operation | Low mountain range | 1205 m | 47°54'49" N, 07°54'29" E |
| 16 | Waldhof | WAL | In operation | Regional background | 75 m | 52°48'04" N, 10°45'23" E |
| 17 | Zugspitze (Schneefernerhaus) | ZSF | In operation | High alpine | 2670 m | 47°25'00" N, 10°58'47" E |




**Table 2: Technical details of GUAN instrumentations. Mobility particle size spectrometers (MPSS) follow the TROPOS design**
**unless stated otherwise (Birmili et al., 2016).**

| NO. | Name | Type | Inlet height above ground | Particle mobility size spectrometer type | Size range | eBC instrument | eBC cut-off |
|---|---|---|---|---|---|---|---|
| 1 | ANA | portable cabin | 4 m | MPSS | 10–800 nm | MAAP | $PM_1$ |
| 2 | AUG | portable cabin | 4 m | D-MPSS | 5–800 nm | Aethalometer (Type 8100) | $PM_{2.5}$ |
| 3 | BOS | portable cabin | 4 m | MPSS | 10–800 nm | MAAP | $PM_{10}$ |
| 4 | DDN | portable cabin | 4 m | D-MPSS | 5–800 nm | MAAP | $PM_1$ |
| 5 | DDW | portable cabin | 4 m | MPSS | 10–800 nm | MAAP | $PM_1$ |
| 6 | HPB | building | 12 m | MPSS | 10–800 nm | MAAP | $PM_{10}$ |
| 7 | LAN | portable cabin | 14 m | MPSS (TSI 3936) | 10–600 nm | – | $PM_1$ |
| 8 | LEI | building | 6 m | TDMPSS | 5–800 nm | MAAP | $PM_1$ |
| 9 | LMI | portable cabin | 4 m | TDMPSS | 5–800 nm | MAAP | $PM_{10}$ |
| 10 | LTR | portable cabin | 16 m | TDMPSS | 5–800 nm | MAAP | $PM_{10}$ |
| 11 | LWE | portable cabin | 4 m | TDMPSS | 10–800 nm | MAAP | $PM_{10}$ |
| 12 | MEL | portable cabin | 4 m | D-MPSS | 5–800 nm | MAAP | $PM_{10}$ |
| 13 | MST | portable cabin | 4 m | MPSS (TSI 3936) | 14–750 nm | – | $PM_{10}$ |
| 14 | NEU | building | 6 m | MPSS | 10–800 nm | MAAP | $PM_{10}$ |
| 15 | SCH | building | 6 m | MPSS | 10–800 nm | MAAP | $PM_{10}$ |
| 16 | WAL | building | 6 m | MPSS | 10–800 nm | MAAP | $PM_{10}$ |
| 17 | ZSF | building | 6 m | MPSS (TSI 3936) | 10–600 nm | MAAP | $PM_{10}$ |






**Table 3: Multi-annual trends of the eBC mass concentration and PNCs in percent per year, using the customized Sen's estimator and generalized linear square regression with autoregression bootstrap (GLS /ARB). The bold slopes are the significant slopes at the 95% significance level. Five site categories on the left column are roadside (RS), urban background (UB), regional background (RB), low mountain range (LMT) and high Alpine (HA).**

| Category | Site | eBC mass concentration | | $N_{[20-800]}$ | | $N_{[10-30]}$ | | $N_{[30-200]}$ | | $N_{[200-800]}$ | |
|---|---|---|---|---|---|---|---|---|---|---|---|
| | | Sen slope | GLS/ARB slope | Sen slope | GLS/ARB slope | Sen slope | GLS/ARB slope | Sen slope | GLS/ARB slope | Sen slope | GLS/ARB slope |
| **RS** | DDN | **-7.7%** | **-8%** | **-5.3%** | **-4.4%** | **-5.7%** | **-5.2%** | **-5.0%** | **-4.3%** | **-6.7%** | **-6.3%** |
| | LEI | **-4.1%** | **-5.1%** | **-3.0%** | **-3.0%** | **-4.1%** | **-3.8%** | **-3.2%** | **-2.9%** | -1.5% | -2.2% |
| | LMI | **-4.4%** | **-5.0%** | **-3.7%** | **-5.5%** | 0.3% | 0.0% | **-4.2%** | **-5.1%** | **-3.9%** | **-5.1%** |
| **UB** | MST | --- | --- | -2.3% | -0.5% | --- | --- | -2.4% | -2.2% | **-4.7%** | **-4.4%** |
| | LTR | **-3.8%** | **-5.0%** | **-3.3%** | **-4.4%** | **-3.6%** | **-5.1%** | **-3.4%** | **-4.1%** | -3.4% | **-6.9%** |
| | ANA | **-5.5%** | **-6.8%** | **-5.0%** | **-5.2%** | **-6.3%** | **-5.4%** | **-4.7%** | **-4.9%** | **-7.7%** | **-5.7%** |
| | AUG | **-2.9%** | **-3.2%** | **-6.7%** | **-5.9%** | **-5.0%** | **-6.3%** | **-7.0%** | **-6.5%** | **-9.5%** | -4.0% |
| | DDW | **-6.3%** | **-7.6%** | **-4.1%** | **-6.5%** | **-3.8%** | **-6.1%** | **-3.9%** | **-6.3%** | **-6.7%** | **-6.4%** |
| | LAN | --- | --- | -3.4% | -2.7% | -0.1% | -0.6% | -3.2% | -3.2% | -4.9% | -0.1% |
| | BOS | **-4.2%** | **-5.1%** | **-4.7%** | **-4.4%** | -1.0% | **-4.6%** | **-4.8%** | **-4.6%** | **-4.7%** | -2.6% |
| **RB** | MEL | **-3.8%** | **-5.5%** | -0.4% | 0.1% | 2.7% | 1.6% | -0.4% | 0.1% | -2.5% | -3.0% |
| | WAL | **-2.9%** | **-3.9%** | **-3.5%** | **-3.0%** | -2.6% | -2.2% | **-3.7%** | **-3.1%** | **-4.3%** | **-4.5%** |
| | NEU | **-5.8%** | **-6.1%** | -0.7% | -0.6% | -1.1% | 0.2% | -0.4% | -0.4% | -3.3% | **-4.4%** |
| **LMT** | HPB | **-2.3%** | **-4.8%** | -1.0% | -2.3% | 1.9% | 0.3% | -0.9% | -0.9% | -3.1% | **-4.7%** |
| | SCH | -1.8% | **-3.5%** | -1.7% | -2.9% | 4.3% | -2.7% | **-2.0%** | **-2.7%** | -3.4% | -3.7% |
| **HA** | ZSF | **-5.0%** | **-7.8%** | **-3.7%** | **-4.4%** | --- | --- | **-3.6%** | **-3.9%** | **-3.8%** | **-7.5%** |
| **Regional Mann-Kendall** | | **-3.8%** | | **-1.6%** | | -1.4% | | **-2.0%** | | **-2.4%** | |







**Table 4: Trend of meteorological parameters all over Germany. The bold numbers are the significant slopes at the 95% significance**
**level. The daily meteorological data are from Germany's National Meteorological Service (Deutscher Wetterdienst, DWD). The**
**mean time series among all 76 sites was used as the area average of meteorological parameters all over Germany.**

| | Precipitation | | | Temperature | | | Wind speed | | |
|---|---|---|---|---|---|---|---|---|---|
| | Slope in mm year$^{-1}$ | CI in mm year$^{-1}$ | | Slope in °C year$^{-1}$ | CI in °C year$^{-1}$ | | Slope in m s$^{-1}$ year$^{-1}$ | CI in m s$^{-1}$ year$^{-1}$ | |
| Spring (MAM) | -0.01 (-0.8%) | -0.07 | 0.05 | -0.04 (-0.4%) | -0.15 | 0.07 | 0.01 (0.2%) | -0.02 | 0.04 |
| Summer (JJA) | **-0.15 (-5.9%)** | -0.22 | -0.06 | **0.15 (0.8%)** | 0.07 | 0.22 | -0.01 (-0.2%) | -0.03 | 0.02 |
| Autumn (SON) | -0.06 (-3.3%) | -0.13 | 0.01 | **0.36 (3.2%)** | 0.32 | 0.50 | -0.03 (-0.6%) | -0.07 | 0.01 |
| Winter (DJF) | 0.04 (1.6%) | -0.03 | 0.12 | **0.41 (11.3%)** | 1.09 | 1.82 | **0.04 (0.7%)** | 0.00 | 0.10 |



**Table 5: Basic statistical information of the different air mass types.**

| Air mass type | Wind direction | Source region | Frequency 2009-2014 (%) | Mean PM$_{10}$ (µg m$^{-3}$) |
|---|---|---|---|---|
| **CS-ST** | Stagnant | Central Europe | 2.6 | 39.6 |
| **CS-A1** | East | Eastern Europe | 4.0 | 36.5 |
| **CS-A2** | West | North Atlantic | 5.6 | 25.6 |
| **CS-C1** | South West | Southwest Europe | 5.2 | 26.6 |
| **CS-C2a** | South West | North Atlantic | 3.6 | 12.8 |
| **CS-C2b** | West | North Atlantic | 5.5 | 13.0 |
| **TS-A1** | North East | Subpolar | 8.3 | 19.8 |
| **TS-A2** | West | North Atlantic | 6.3 | 18.7 |
| **TS-C1** | South West | Southwest Europe | 5.1 | 15.4 |
| **TS-C2** | North West | Arctic | 10.8 | 14.1 |
| **WS-ST** | Stagnant | Central Europe | 6.8 | 23.2 |
| **WS-A1** | South East | Eastern Europe | 5.6 | 28.4 |
| **WS-A2** | North West | North Atlantic | 12.4 | 17.9 |
| **WS-C1** | West | North Atlantic | 9.7 | 18.0 |
| **WS-C2** | West | North Atlantic | 8.3 | 13.0 |





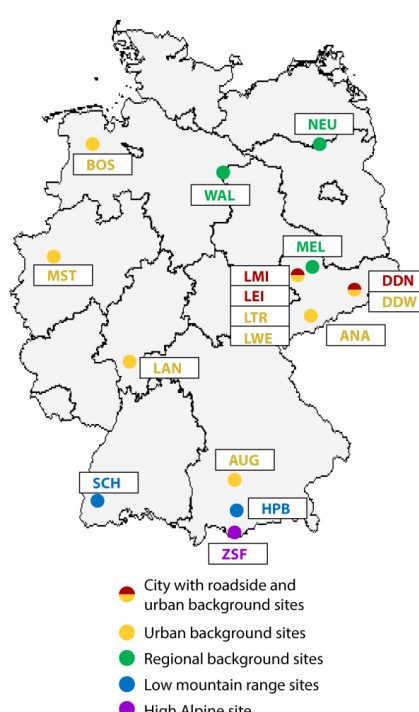

| Abbreviation | Site name |
|---|---|
| ANA | Annaberg-Buchholz |
| AUG | Augsburg |
| BOS | Bösel |
| DDN | Dresden-Nord |
| DDW | Dresden-Winckelmannstraße |
| HPB | Hohenpeißenberg |
| LAN | Langen |
| LEI | Leipzig-Eisenbahnstraße |
| LMI | Leipzig-Mitte |
| LTR | Leipzig-TROPOS |
| LWE | Leipzig-West |
| MEL | Melpitz |
| MST | Mülheim-Styrum |
| NEU | Neuglobsow |
| SCH | Schauinsland |
| WAL | Waldhof |
| ZSF | Zugspitze (Schneefernerhaus) |

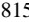


**Figure 1: The map of atmospheric measurement stations in GUAN.**


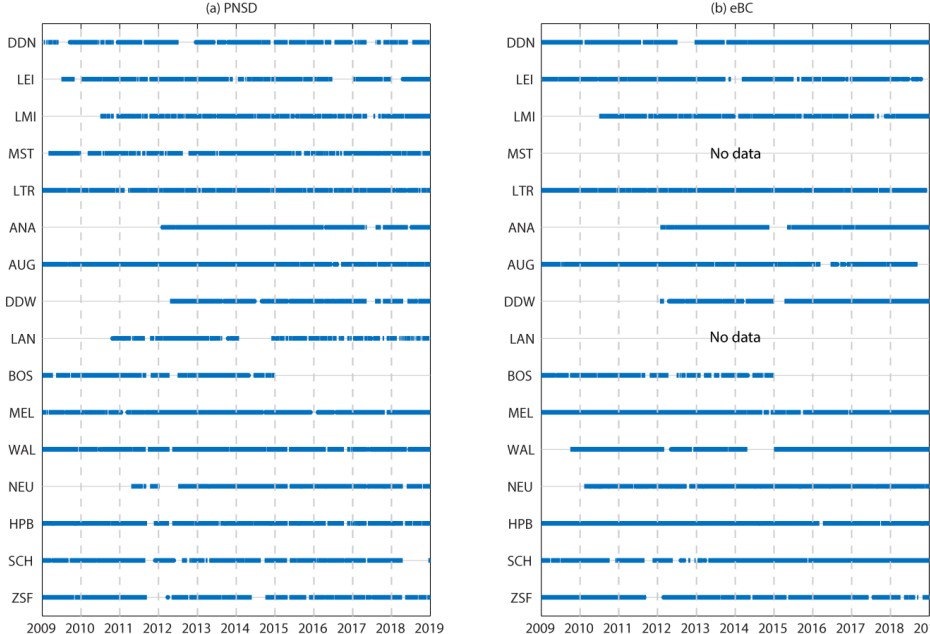


**Figure 2: Data coverage of the PNSD and the eBC mass concentration at GUAN sites, from 2009 to 2018.**




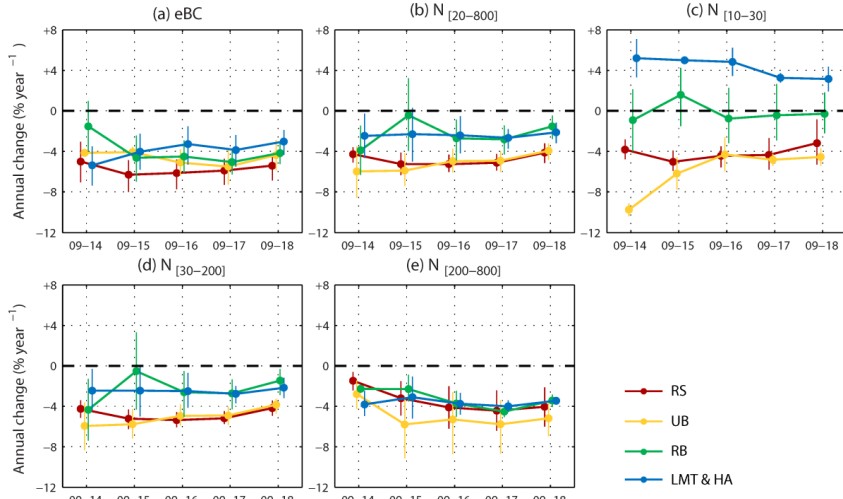


**Figure 3: Annual trends of the eBC mass concentration and PNCs for expanding time intervals starting from 2009, using the customized Sen's estimator. The x-axis shows the starting and ending year of each data point. The dot indicates the mean slope and the whiskers denote the 75th and 25th percentiles. The trend evolution for each site category is illustrated: roadside (RS), urban background (UB), regional background (RB), low mountain range and high Alpine (LMT&HA).**



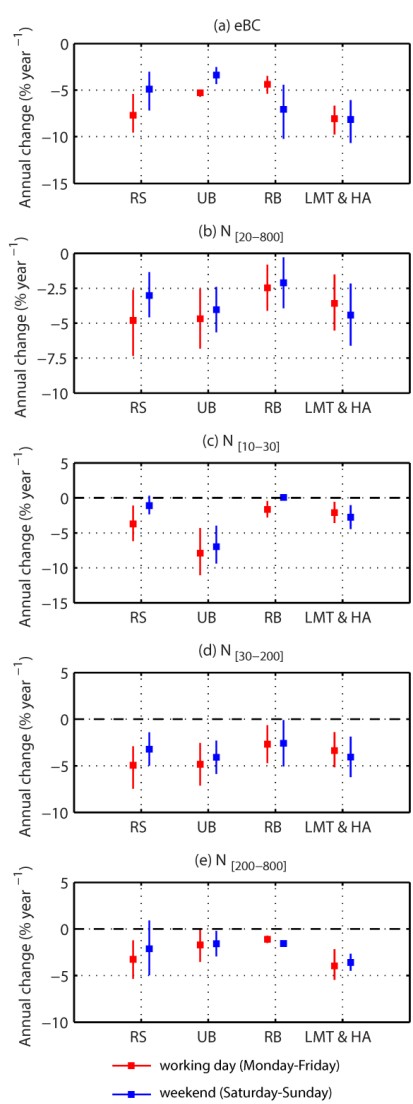

824

**Figure 4: Annual trends of the eBC mass concentration and PNCs for working days and weekend, using the customized Sen's estimator at each site category: roadside (RS), urban background (UB), regional background (RB), low mountain range and high Alpine (LMT&HA). The square denotes the average Sen's slope on corresponding days (working day or weekend) and the whiskers denote the 25th and 75th percentile of Sen's slopes.**




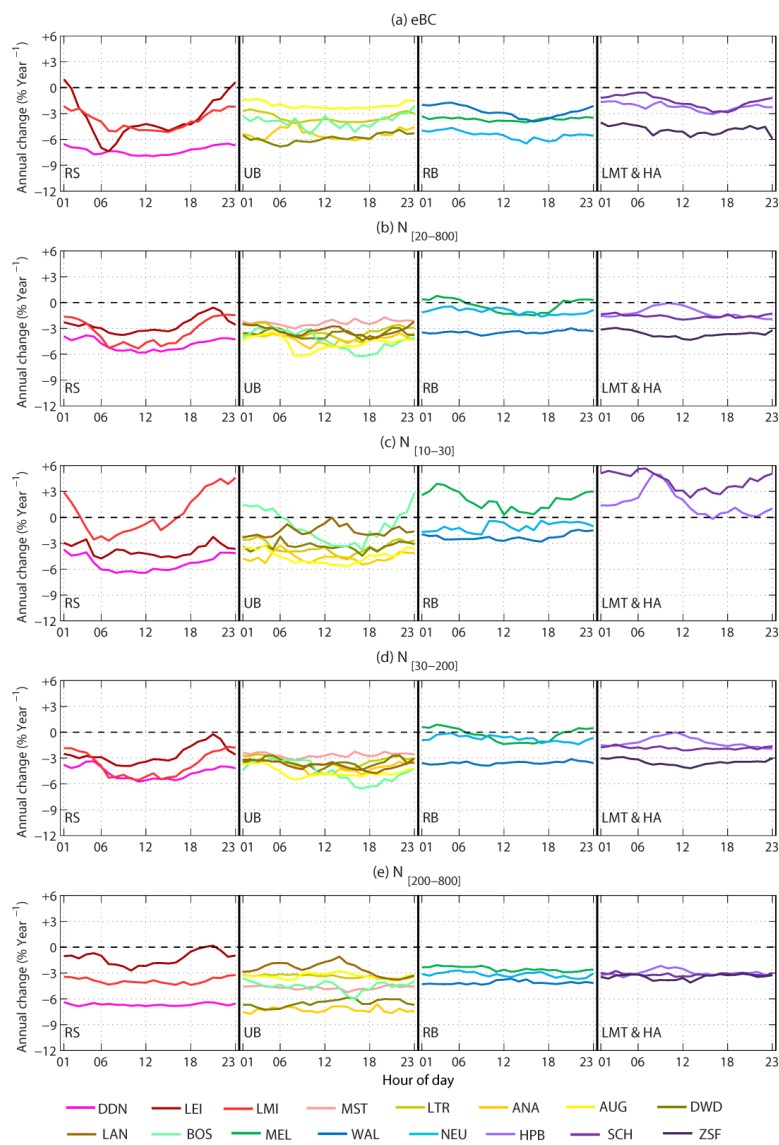


**Figure 5:** Multi-annual trends of the eBC mass concentration and PNCs corresponding to each hour of day, based on the customized Sen's estimator.






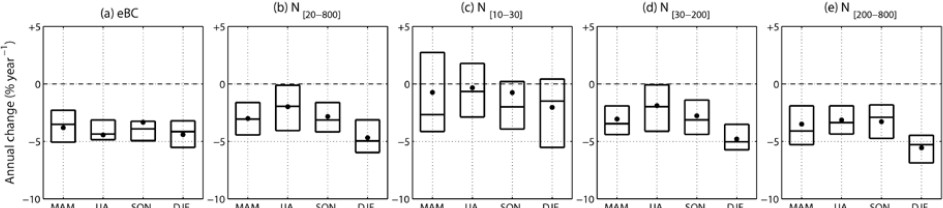


**Figure 6: Seasonal statistics of annual trends of the eBC mass concentration and PNCs, based on the customized Sen's estimator: Spring: March to May (MAM); summer: June to August (JJA); autumn: September to November (SON) and winter: December to February (DJF). Dots refer to mean slope at all sites, black line inside the box refers to the median slope, the top and bottom of box denotes the 75th and 25th percentiles.**

841

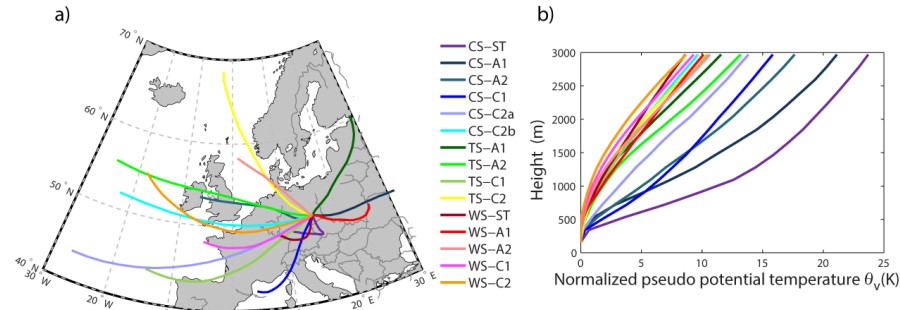

842

**Figure 7: Basic information on the back-trajectory cluster model (BCLM). a): 15 back-trajectory cluster centers terminated at MEL as an example. The duration of the back trajectories is 72h. The name of each air mass cluster refers to the character of each cluster: CS: cold season; TS: transition season; WS: warm season; ST: Stagnant; A: Anticyclonic; C: Cyclonic. b) Average normalized profiles of pseudo potential temperature ($\theta_v$) for the 15 air mass clusters. Profiles with a flat gradient indicate temperature inversions, while a steep gradient, imply stratification close to neutral. Data originate from the radiosounding launched at the DWD station Lindenberg, located 115 km northeast of MEL.**

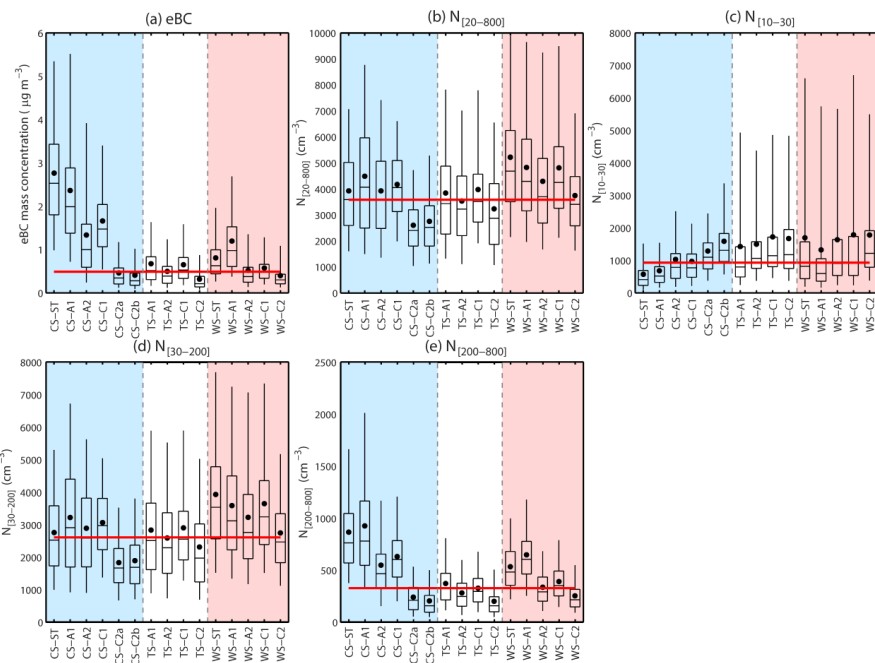


**Figure 8: Average concentration value of eBC mass concentration (a) and size-dependent PNCs (b to e) for the 15 air mass types at**
**regional background site category (Sites MEL, WAL and NEU). For each panel, the boxes and whiskers denote the 5th, 25th, 50th,**
**75th and 95th percentiles, while the dots denote the mean values. The solid red line indicates the overall median values.**

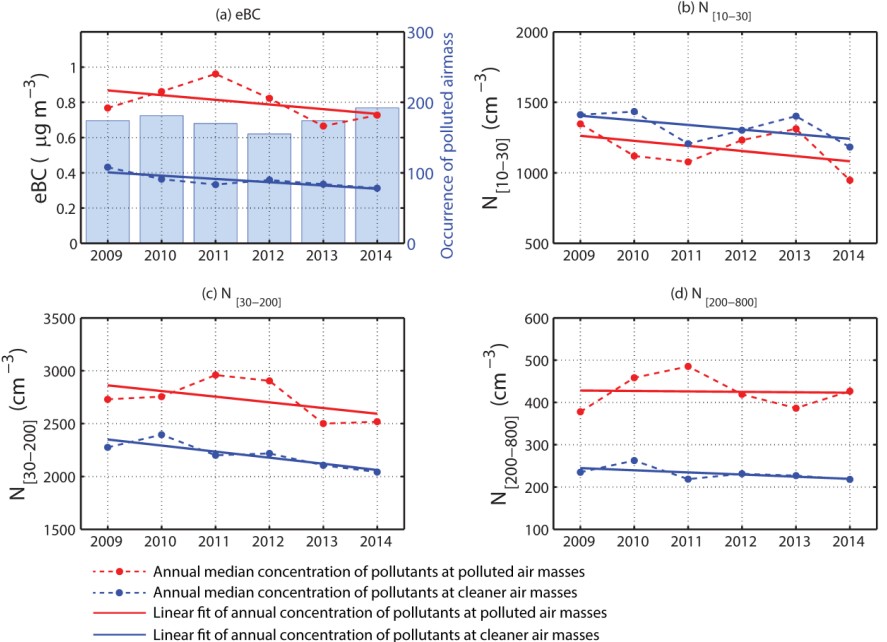


**Figure 9: Annual concentration of the eBC mass concentration and PNCs for the two air mass categories, and frequency of polluted**
**air masses.**