# Peer review of "Decreasing Trends of Particle Number and Black Carbon Mass"

_Atmospheric Chemistry and Physics, 2019_

## Referee Comment (RC1) · Anonymous Referee #1 · 30 Sep 2019

The authors reported historical trends of particle number and black carbon concentrations. The article is well-written. I recommend to publish in ACP. A major comment is given hereïïjŽ The authors try to link the pollutants historical trends with mitigation policies. However, only few pollution control regulations are introduced in the introduction section. These regulations should be also mentioned to explain the changes in pollutant concentrations. Thus, it would be much better for understanding the long-term effects of emission mitigation policies. Some implications on the benefit of decreasing PN and BC, for examples, health effects and visibility, should be added at the end of article to enhance the scientific and policy significance.

---

## Referee Comment (RC2) · Anonymous Referee #2 · 15 Oct 2019

The paper presents a valuable dataset and analyses long-term trends of PNC and BC across Germany. The topic is within the scope of ACP, but in my opinion the scientific question does not have to be only how is it changing but why is that as well. Some of the presented conclusions could be better supported by data. For example, if the driver of the PNC and BC decrease is expected to be an emission decrease, could this be compared to any reported emission data? Or any mitigation strategies results? Comparison of the trends with such data would be an added value to the manuscript. Quite some information on BC has been already published in a more detailed paper by Kutzner et al, with more stations and longer dataset. It would be good to include an explanation on what this manuscript brings in addition to the already published results?

[Figure]

The methods and data quality are mostly appropriate with some exceptions. The number of evaluated stations is changing during the text. Why there are stations with no or non-analysed data? If LWE is not evaluated, why is it included in the text? If LAN and Raunheim stations are used for some analyses, why have not these been used from the beginning? At L278, 5 parameters at 16 sites makes 80 trends, why only 77 of them was evaluated? For the PNC data description, the uncertainty of the PNC measurements could be discussed in the text a bit more, (L164 etc.) and compared to the presented trends. In the 5.1 Section, a mean value of meteorological parameters is used for all stations. Would not it be better to have at least three different averages for the different types of stations? It would be difficult to compare one T and RH value for Alpine site, city etc. (L416) Also the 5.2 section needs more detailed methodology description. Why 15 clusters were used, what data were used for trajectory calculation? And mainly, why the analyses have not been done for the whole period? No changes in the period 2009-2014 do not automatically mean there will be no changes in 2009-2018 as well (L444). Also it is not described what is the difference between for example A1 and A2 cluster?

Minor comments: The manuscript would definitely profit from a native speaker check, there are multiple not very usual English phrases – L60 early regions (first?), L343 declined emissions => decreased?, L493+506 downward trend => decreasing trend?, L330+331 LENGTH of the time series - sometimes a verb is missing (L277 monthly median time series WERE USED?, L282 only MEL SHOWS increase?) or mismatched (L355 there is no difference can be seen between, L440 shown => showed?), there is a superfluous use of commas, for example as "it should be noted that, three sites... L300, L312,335 etc", and also some minor typos, for example L316 concentrations are in consistent, L321, Po Valey, one large industrial district.

In the equation description in 2.3.2 and 2.3.3. sections, the symbols are not clearly described, so it is quite difficult to follow the methodology. For example, Eq. 3 does not say what the 2 pi t or 4 pi t means etc. on the other end, it is not necessary to show

how a general vector or matrix looks like, L219 to 221 (with vector missing the C). If the description would be less technical and more explaining what is what, it would be easier to follow. Similarly, the theta in Eq. 9 is not explained at all, and although I know how the signum function works, I cannot recognize it in the eq?

-L363+368 eBC instead of BC?

-L72-73 sentence does not fit either to the preceding or the following sentences

-What does it mean "dataset is sufficient for true slopes"? L336

-L366 higher reduction rate is observed when human activities are more intensive?

-Why is the N10-30 described as "influenced by NPF" called young Aitken and not nucleation as usual?

-Number of references at some sections is redundant, for example 8 references stating non-parametric test are used for trend analysis? L183

-L398 do you expect the biogenic emissions in summer to have a trend? If not, they should not mask the anthropogenic trend?

-L526 the a) explanation does not explain anything, it just repeats the previous sentence?

---

## Author Comment (AC1) · 6 Mar 2020

**Response to comments of referee #1**

**General comments:**

The authors reported historical trends of particle number and black carbon concentrations. The article is well-written. I recommend to publish in ACP. A major comment is given here. The authors try to link the pollutants historical trends with mitigation policies. However, only few pollution control regulations are introduced in the introduction section. These regulations should be also mentioned to explain the changes in pollutant concentrations. Thus, it would be much better for understanding the long-term effects of emission mitigation policies. Some implications on the benefit of decreasing PN and BC, for examples, health effects and visibility, should be added at the end of article to enhance the scientific and policy significance.

Response:

Many thanks for the comments and suggestions.

Following your suggestion of "regulations should be also mentioned to explain the changes in pollutant concentrations" and the general comment 1 of the referee #2, we have rewritten most part of the result section of the manuscript, trying to find more connections between the observed trends and emission variations. A new section "**3.2 Emission change in Germany**" has been added in the manuscript, in which the overall trends of eBC mass concentration and PNCs are compared with the emission data reported by the Federal Environment Agency. Section "**3.3 Diurnal variation of trends**" and "**3.4 Seasonal variation of trends**" have been rewritten and we tried to find the connections between the diurnal and seasonal trends of observed parameters and the sources which have also distinct diurnal or seasonal variations. As a special case study, a new section "**3.5 Evaluation of low emission zones**" has been added in the manuscript to figure out if such a long-term observation network can reflect the effect of a specific emission mitigation policy.

Following your suggestion "Some implications on the benefit of decreasing PN and BC, for examples, health effects and visibility, should be added at the end of article to enhance the scientific and policy significance", a short discussion has been added at the end of Sect. 3.2:

"Based on the above results, we believe that the observed trends of PNCs and eBC mass concentration are mainly due to the reduction in emissions. The annual changes of meteorological conditions might have an impact on PNCs, but are not likely to be the decisive impact factor. Detailed discussion on the possible influence of meteorological conditions will be discussed in Sect. 4. The decreased pollutant concentrations are highly associated with the reduced risk of human health. Pope et al. (2009) demonstrated that a decrease of 10 $\mu$g m$^{-3}$ in the PM$_{2.5}$ mass concentration is related with an increase of life expectancy of $0.61 \pm 0.20$ year in 211 countries. The improved health effects because of decreased UFP and BC would be even greater compared with that of PM$_{2.5}$ mass concentration. As of 2018, 97 % of cities in low- and mid-income countries do not meet the World Health Organization (WHO) air quality guidelines (WHO, 2018). Our result demonstrates that the implementation of proper emission mitigation policies can largely reduce the BC mass concentration and PNC, thus may effectively reduce the health risk in polluted regions."

---

## Author Comment (AC2) · 6 Mar 2020

**Response to comments of referee #2**

**General comments:**

(1) The paper presents a valuable dataset and analyses long-term trends of PNC and BC across Germany. The topic is within the scope of ACP, but in my opinion the scientific question does not have to be only how is it changing but why is that as well. Some of the presented conclusions could be better supported by data. For example, if the driver of the PNC and BC decrease is expected to be an emission decrease, could this be compared to any reported emission data? Or any mitigation strategies results? Comparison of the trends with such data would be an added value to the manuscript.

Response:

Thanks for your suggestions. We have rewritten most part of the result section in the manuscript. A new section "**3.2 Emission change in Germany**" has been added in the manuscript, in which the overall trends of eBC mass concentration and PNCs are compared with the emission data reported by the Federal Environment Agency (Fig. 3 in the manuscript). We found that total emission of BC in Germany deceases about −3.4 % per year during 2009−2017 and highly agrees with the trend of observed eBC mass concentration, suggesting that emission reduction is very likely to be the dominant factor for the decrease in eBC mass concentration over Germany. The total emission of PM$_{2.5}$ and precursors show decreasing trends as well. However, the decreases in PNCs are stronger. This discrepancy is thought to be caused by the highly complex and nonlinear processes of secondary aerosol formation. Based on the comparisons, we believe that the observed trends of eBC mass concentration and PNCs are mainly due to the emission reduction.

[Figure]

Figure 3: Comparison of the long-term changes in measured parameters and total emissions in Germany.

The emission intensities of some sources have distinct diurnal or seasonal variations, such as that of traffic and residential activities. Section "**3.3 Diurnal variation of trends**" and "**3.4 Seasonal variation of trends**" have been rewritten and we tried to find out the connections between the diurnal and seasonal trends of observed parameters and those sources.

As a special case study, a new section "**3.5 Evaluation of low emission zones**" has been added in the manuscript. The reason for adding this section is, the observed decrease in trends is usually a combined result of the various emission mitigation policies. A question raised is that can such long-term observation network reflect the effect of a specific emission mitigation policy. We select one special mitigation policy low emission zone (LEZ) and trying to figure out its effectiveness based on our dataset. We find gradual decreases in eBC mass concentration and $N_{[30-200]}$ after the implementation of LEZ. But even with the seasonal variation subtracted from the time series, the amplitude of variation of eBC mass concentration and $N_{[30-200]}$ is still very large mostly due to variations in meteorological conditions (Fig. 6 in the manuscript). However, very clear responses are found in the increment of the aerosol concentration (the difference of the concentrations between the traffic and the background sites) (Fig. 7 in the manuscript), suggesting that with a multiple-site network, the effect of emission control policy could be better detected from the increments between near-source and background sites.

[Figure]

Figure 6: De-seasonalised monthly time series of eBC mass concentration and $N_{[30-200]}$ at the two urban background sites AUG and LTR. The vertical dashed lines refer to the start dates of LEZ of different stages in the city of Augsburg and Leipzig. The horizontal dashed lines refer to the mean concentration levels of measured parameters during the corresponding time period.

[Figure]

Figure 7: Average diurnal cycles of the increment (defined as the difference between LMI and LTR) in eBC mass concentrations, $N_{[30-200]}$ and $N_{[200-800]}$.

(2) Quite some information on BC has been already published in a more detailed paper by Kutzner et al, with more stations and longer dataset. It would be good to include an explanation on what this manuscript brings in addition to the already published results?

Response:

Thanks for the suggestion. There are several differences between Kutzner et al. (2018) and our study.

(a) Data from 12 stations in 3 states were used in Kutzner et al. (2018) for the trend analysis. In our study, 16 stations distributed in 8 states (Fig. 1 in the manuscript) were used and therefore can better represent the variation of BC concentration across Germany.

(b) Most of the stations (11 out of 12) are located in urban area and largely influence by traffic. Thus, only the mitigation of traffic emission was considered in explaining the trend of BC in Kutzner et al. (2018). In our study, stations in various environments (3 roadside, 5 urban background, 3 regional background, and 3 mountain area) are used, and the changes of different sources (traffic, domestic heating, other fuel combustions, industry, etc.) were considered.

(c) Only general trend of BC is reported in Kutzner et al. (2018). In our study, the diurnal and seasonal variations in the long-term trend of BC were investigated and connected to the change of specific emissions.

(d) A comprehensive evaluation of the influence of the inter-annual variations of meteorological conditions (e.g., air masses, precipitation and temperature) is presented in our study.

Following your suggestions, the following sentences have been added at the end of Sect. 1 in the manuscript:

"Kutzner et al. (2018) evaluated the trend of BC over Germany based on measurement at traffic, urban background, and rural sites for the period of 2005–2014, and concluded that the observed decreasing trends in BC are likely owing largely to mitigation measures in the traffic sector. However, there is still a lack of a thorough investigation of the connections between the long-term trends of PNCs/BC and the change of different anthropogenic emissions. A better understanding of the influence of the inter-annual variation of meteorological conditions on the observed trends is also needed.

Based on a unique dataset from the German Ultrafine Aerosol Network (GUAN), this study investigates the long-term variation in the regional PNC and BC mass concentration, to understand the effectiveness of the emission mitigation policies in reducing the PNC and BC in Germany. The study was conducted for the period of 2009–2018 with data from 16 observational sites representing different types of environment (roadside, urban background, regional background, low mountain, and high Alpine). The overall, diurnal, and seasonal trends of PNCs and BC are evaluated and the role of potential decisive factors, including not only emission mitigation policies, but also other potential drivers (i.e. inter-annual change in meteorological conditions and long-range transport patterns) are discussed."

Moreover, a comparison between our results and other studies is given in Sect. 3.1:

"The trends of the PNCs and eBC mass concentration in this study are consistent with results from other studies conducted in Europe. Table 4 compares the long-term trends of aerosol concentrations between the present and other studies. Since 2001, the s.s. decrease in BC, PNCs, and PM$_{2.5}$ have been detected for most of the evaluated sites in Table 4. The implementation of emission mitigation polices have been thought to be the dominant impact factors in these studies. Especially, there is one similar study that evaluated the trend of BC mass concentration in Germany for the time period 2005−2014 (Kutzner et al., 2018), in which decreased BC mass concentration was detected in 12 sites. Comparing the two studies, the absolute decreasing trend of BC mass for 2005-2014 is stronger than our results for 2009−2018, which might stem from the difference between the effects of emission mitigation policies in the two study periods."

Table 4. Comparison of long-term trend studies of BC, PNC, and PM in Europe.

| Study | Time period | Region | Parameters | Annual slope (numbers in brackets are the absolute slope, in µg m$^{-3}$ year$^{-1}$) |
|---|---|---|---|---|
| This study | 2009−2018 | Germany | BC | Traffic (3 sites): −11.3 %~−5.0 %, (−0.19~−0.08); UB (5 sites): −8.1 %~−2.3 % (−0.08~−0.03); RB to high Alpine (6 sites): −7.8 %~−1.7 % (−0.03~0.00) |
| | | | $N_{[20–800]}$ | Traffic (3 sites): −7.3 %~−2.9 %; UB (7 sites): −6.3 %~−2.6 %; RB to high Alpine (6 sites): −4.2 %~−0.2 % |
| Kutzner et al., 2018 | 2005−2014 | Germany | BC | Traffic (7 sites): (−0.31, −0.15); UB (4 sites): (−0.1, −0.02); Rural (1 site): 0.00 |
| Asmi et al., 2013 | 2001−2010 | Europe | $N_{[20–800]}$ | Rural to remote (4 sites): −4.6 %~1.6 % |
| Collaud Coen et al., 2013 | 2001−2010 | Europe | Absorption coef. | Rural to remote (4 sites): −1.6 %~0.0 % |
| Bigi and Ghermandi, 2016 | 2005−2014 | Italy, Po valley | PM$_{2.5}$ | Traffic (2 sites): −6.4 %~−4.6 %; UB (17 sites): −8.1 %~ −0.4 %; RB (4 sites): −4.9 %~0.0 % |
| Singh et al., 2018 | 2009−2016 | United Kingdom | BC | Traffic (1 site): −8.0 %; UB (2 sites): −5.0 %~−4.7 %; Rural (1 site): −7.7 % |

(3) The methods and data quality are mostly appropriate with some exceptions. The number of evaluated stations is changing during the text. Why there are stations with no or non-analysed data? If LWE is not evaluated, why is it included in the text? If LAN and Raunheim stations are used for some analyses, why have not these been used from the beginning? At L278, 5 parameters at 16 sites makes 80 trends, why only 77 of them was evaluated?

Response:

Thanks for the comment. The German Ultrafine Aerosol Network (GUAN) includes 17 stations. However, one station Leipzig-West (LWE) has been terminated in 2016 and it shows high similarity as site Leipzig-TROPOS (LTR). Therefore, we decided not to include LWE in this study. Only 16 sites were used. To avoid confusion, we have deleted LWE in site description and in Fig.1, table 1 and 2 in the manuscript.

At site LAN, there is no measurement of eBC mass concentration. Thus, we decided to use another site Raunheim which is not a GUAN site but close to LAN, to help detect the BC trend. In the revised manuscript, we have deleted the results from Raunheim to avoid confusion.

For the question about the number of trends, as shown in Fig. 2 in the manuscript, the eBC mass concentrations were not measured at MST and LAN. PNSD measurements at MST and ZSF start from diameter of 14 and 20 nm, respectively. Thus no information of $N_{[10\text{-}30]}$ is available at these two sites. Therefore, for 16 sites and five parameters, only 76 trends in total were evaluated. To make it clear, a new table (Table S1) has been added in the supplemental material giving the number of analysed stations with respect to all five parameters, and the sentence in Sect. 3.1 has been revised as "For the five parameters at the 16 sites (76 trends in total, see Table S1 in SM), the trends…"

Table S1: Number of sites used in trend analysis.

| Parameters | Number of stations analysed | Excluded stations |
|---|---|---|
| eBC | 14 | MST, LAN |
| $N_{[20-800]}$ | 16 | -- |
| $N_{[10-30]}$ | 14 | MST, ZSF |
| $N_{[30-200]}$ | 16 | -- |
| $N_{[200-800]}$ | 16 | -- |

(4) For the PNC data description, the uncertainty of the PNC measurements could be discussed in the text a bit more, (L164 etc.) and compared to the presented trends.

Response:

Thanks for your comment. The following paragraph has been added in Sect. 2.2 in the text.

"Quality assurance of PNSD measurements in GUAN are periodically done to ensure that measurements remain stable both instrument to instrument (or site to site) and instrument to standard. Monthly maintenance and onsite/laboratory inter-comparisons with a reference MPSS with a frequency between one to four times per year as recommended by Wiedensohler et al. (2018) are done by the World Calibration Centre for Aerosol Physics (WCCAP, http://www.wmo-gaw-wcc-aerosol-physics.org/). These procedures can ensure an accuracy of ±10 % for PNCs over the entire measurement period (Birmili et al., 2016). Although the uncertainty of PNCs is comparable or higher than their annual trends (Sect. 3.1), with the application of periodical quality assurance procedures, there should be no monotonicity change or

systematic bias in the measurement uncertainties. Therefore, the influence of the measurement uncertainty on the detection of long-term trends of PNCs is assumed to be negligible."

(5) In the 5.1 Section, a mean value of meteorological parameters is used for all stations. Would not it be better to have at least three different averages for the different types of stations? It would be difficult to compare one T and RH value for Alpine site, city etc. (L416)

Response:

   Thanks for the comments. Following your suggestion, the 76 DWD stations were classified into three categories (Table 5 in the manuscript): urban background, regional background and mountain area. And section "4.1 Influence of precipitation, temperature, wind speed on the detected trends" has been revised accordingly.

Table 5: Trends of meteorological parameters for the three site categories in Germany. The bold numbers are the statistically significant slopes at the 95 % significance level. The daily meteorological data are from Germany's National Meteorological Service (Deutscher Wetterdienst, DWD).

| season | | Urban background | Regional background | Mountain area |
|---|---|---|---|---|
| Spring | Precipitation mm year$^{-1}$ (% year$^{-1}$) | −0.02 (−1.0) | 0.00 (0.0) | −0.02 (−0.5) |
| | Temperature °C year$^{-1}$ | −0.04 | −0.03 | −0.02 |
| | Wind speed m s$^{-1}$ year$^{-1}$ (% year$^{-1}$) | 0.01 (0.2) | 0.02 (0.3) | 0.04 (0.6) |
| Summer | Precipitation mm year$^{-1}$ (% year$^{-1}$) | **−0.14 (−5.5)** | **−0.15 (−5.8)** | **−0.20 (−4.7)** |
| | Temperature °C year$^{-1}$ | **0.15** | **0.13** | **0.16** |
| | Wind speed m s$^{-1}$ year$^{-1}$ (% year$^{-1}$) | 0.00 (0.0) | 0.02 (0.4) | **−0.08 (−1.4)** |
| Autumn | Precipitation mm year$^{-1}$ (% year$^{-1}$) | −0.07 (−3.9) | −0.05 (−2.5) | **−0.07 (−1.9)** |
| | Temperature °C year$^{-1}$ | **0.37** | **0.36** | **0.29** |
| | Wind speed m s$^{-1}$ year$^{-1}$ (% year$^{-1}$) | −0.02 (−0.8) | −0.01 (−0.3) | **−0.09 (−1.2)** |
| Winter | Precipitation mm year$^{-1}$ (% year$^{-1}$) | 0.02 (1.3) | 0.04 (1.8) | **0.14 (3.1)** |
| | Temperature °C year$^{-1}$ | **0.41** | **0.43** | **0.34** |
| | Wind speed m s$^{-1}$ year$^{-1}$ (% year$^{-1}$) | 0.02 (0.5) | 0.05 (0.9) | **0.13 (1.5)** |

(6) Also the 5.2 section needs more detailed methodology description. Why 15 clusters were used, what data were used for trajectory calculation? And mainly, why the analyses have not been done for the whole

period? No changes in the period 2009-2014 do not automatically mean there will be no changes in 2009-2018 as well (L444). also it is not described what is the difference between for example A1 and A2 cluster?

Response:

Thanks for the comment. Following your suggestion, we have added a more detailed description in supplementary information (Sect. 3 in the supplemental material) about the back-trajectory cluster method, including the basic description, data sources, cluster algorithm, and the evaluation of cluster results.

Guided by experience from previous studies (Engler et al., 2012; Ma et al., 2014), we tested the cluster algorithm for a range of cluster numbers $k$ between 8 and 19. The best solution was obtained with cluster number 15. More and more redundancies in the cluster composition (i.e. cluster means close to each other) were observed for $k > 15$; while reducing the number of clusters below 15 would, conversely, merge clusters that could be clearly identified as different typical weather situations in Central Europe.

The trajectories were calculated using a PC version of HYSPLIT (Stein et al., 2015) with Global Data Assimilation System (GDAS) analysis set which provides meteorological fields every 3 hours, at a horizontal resolution of 1°, and at numerous standard pressure levels. In our Back Trajectory and Temperature Profile (BTTP) cluster method, vertical profiles of pseudopotential temperature $\theta_v$ retrieved from radiosounding data at seven stations are also used for the classification of trajectories.

We have extended the clustering for the whole time period (2009−2018) in the revised manuscript. The corresponding figures, tables and text have been updated. The new results are similar as that for the period of 2009−2014.

For the comment "also it is not described what is the difference between for example A1 and A2 cluster?", following description of the 15 air mass types has been added in Sect. 4.2 in the text.

"The 15 air mass types are named by seasons (CS: cold season; TS: transition season; and WS: warm season) and synoptic patterns (ST: Stagnant; A1: Anti-cyclonic with air mass originating from Eastern Europe; A2: Anti-cyclonic with air mass originating from west; C1: cyclonic with air mass originating from relatively south; C2: cyclonic with air mass originating from the north). Table 6 lists the basic statistical information of the 15 air mass types."

**Minor comments:**

(1) The manuscript would definitely profit from a native speaker check, there are multiple not very usual English phrases –

L60 early regions (first?),

L343 declined emissions => decreased?,

L493+506 downward trend => decreasing trend?

L330+331 LENGTH of the time series - sometimes a verb is missing (L277 monthly median time series WERE USED?, L282 only MEL SHOWS increase?) or mismatched (L355 there is no difference can be seen between, L440 shown => showed?)

there is a superfluous use of commas, for example as "it should be noted that, three sites: L300, L312,335 etc",

also some minor typos, for example L316 concentrations are in consistent, L321, Po Valey, one large industrial district.

Response:

Many thanks for the corrections and suggestions. Most parts of the manuscript have been rewritten, and above sentences have been corrected or deleted. The manuscript has been also edited by Elsevier Language Editing Services to improve the language.

(2) In the equation description in 2.3.2 and 2.3.3. sections, the symbols are not clearly described, so it is quite difficult to follow the methodology. For example, Eq. 3 does not say what the 2 pi t or 4 pi t means etc. on the other end, it is not necessary to show how a general vector or matrix looks like, L219 to 221 (with vector missing the C). If the description would be less technical and more explaining what is what, it would be easier to follow. Similarly, the theta in Eq. 9 is not explained at all, and although I know how the signum function works, I cannot recognize it in the eq?

Response:

Thanks for the comment. The GLS/ARB is a well-developed method and adapted by Asmi et al. (2013) for trend analysis of aerosol concentrations. Since we use exactly the same method as that in Asmi et al. (2013) and all details can be found in the book by Mudelsee (2010) and the paper by Asmi et al. (2013), we decided not to repeat the details in the main text, to make the text shorter and easy to read. Section "**2.3.2 Generalized least-square-regression and auto-regressive bootstrap confidence intervals**" and "**2.3.3 Regional Mann-Kendall test**" has been shortened accordingly.

(3) L363+368 eBC instead of BC?

Response:

Thanks. We have rewritten the entire paragraph and this sentence has been removed. In the revised manuscript, the term "BC" is used to stand for aerosol species black carbon, and the term "eBC" is used to stand for the measurement data of MAAP and Aethalometer.

(4) L72-73 sentence does not fit either to the preceding or the following sentences

Response:

The sentences have been revised as "For domestic heating emission, the unsuitable fuels are listed and their emission values are defined. For traffic emission, low emission zones (LEZs) were set up to limit the emission of nitrogen oxide and aerosol particles from traffic exhaust."

(5) What does it mean "dataset is sufficient for true slopes"? L336

Response:

The section has been moved to supplementary information. And the sentence in the main text has been revised as "Robustness analysis (see Sect. 1 in the supplementary material) suggests that the time span of

our dataset is long enough for slope detection and that the influence of measurement uncertainty is negligible."

(6) L366 higher reduction rate is observed when human activities are more intensive?

Response:

Here, "human activities" means traffic, domestic heating, cooking etc. This section has been rewritten and the corresponding sentence has been removed.

(7) Why is the N10-30 described as "influenced by NPF" called young Aitken and not nucleation as usual?

Response:

The term "young Aitken mode" has been replaced by "nucleation mode" in the revised manuscript.

(8) Number of references at some sections is redundant, for example 8 references stating non-parametric test are used for trend analysis? L183

Response:

Thanks. Some of the references have been removed in the manuscript.

(9) L398 do you expect the biogenic emissions in summer to have a trend? If not, they should not mask the anthropogenic trend?

Response:

We do not except any trend in the biogenic emission in summer. Biogenic emissions contribute considerable SOA precursors and thus a large contribution on PNCs. Therefore, the relative contribution of anthropogenic emission is lower in summer than in other seasons. Without any strong long-term variation in biogenic emission (means a large portion in PNC does not change), its stable contribution on PNCs may lower the relative decreasing rates in PNCs in summer. To make it clearer, the sentence has been revised as:

"Other than the low residential emission in warm seasons, another reason might be the strong seasonal variations in biogenic emissions (Asmi et al., 2013). Biogenic emissions contribute considerable secondary organic aerosol (SOA) precursors in summer and thus a higher contribution on PNCs. Without any strong long-term variation, the stable contribution of biogenic emissions on PNCs might lower the relative decreasing rates in PNCs in summer."

(10) L526 the a) explanation does not explain anything, it just repeats the previous sentence?

Response:

Thanks. We have rewritten the entire paragraph and this sentence has been removed.

**Reference**

Asmi, A., Collaud Coen, M., Ogren, J. A., Andrews, E., Sheridan, P., Jefferson, A., Weingartner, E., Baltensperger, U., Bukowiecki, N., Lihavainen, H., Kivekas, N., Asmi, E., Aalto, P. P., Kulmala, M., Wiedensohler, A., Birmili, W., Hamed, A., O'Dowd, C., Jennings, S. G., Weller, R., Flentje, H., Fjaeraa, A. M., Fiebig, M., Myhre, C. L., Hallar, A. G., Swietlicki, E., Kristensson, A., and Laj, P.: Aerosol decadal trends - Part 2: In-situ aerosol particle number concentrations at GAW and ACTRIS stations, Atmos. Chem. Phys., 13, 895-916, 10.5194/acp-13-895-2013, 2013.

Engler, C., Birmili, W., Spindler, G., and Wiedensohler, A.: Analysis of exceedances in the daily $PM_{10}$ mass concentration (50 µg m$^{-3}$) at a roadside station in Leipzig, Germany, Atmos. Chem. Phys., 12, 10107-10123, 10.5194/acp-12-10107-2012, 2012.

Kutzner, R. D., von Schneidemesser, E., Kuik, F., Quedenau, J., Weatherhead, E. C., and Schmale, J.: Long-term monitoring of black carbon across Germany, Atmos. Environ., 185, 41-52, https://doi.org/10.1016/j.atmosenv.2018.04.039, 2018.

Ma, N., Birmili, W., Müller, T., Tuch, T., Cheng, Y. F., Xu, W. Y., Zhao, C. S., and Wiedensohler, A.: Tropospheric aerosol scattering and absorption over central Europe: a closure study for the dry particle state, Atmos. Chem. Phys., 14, 6241-6259, 10.5194/acp-14-6241-2014, 2014.

Mudelsee, M.: Climate Time Series Analysis: Classical Statistical and Bootstrap Methods., Springer, 2010.

Stein, A. F., Draxler, R. R., Rolph, G. D., Stunder, B. J. B., Cohen, M. D., and Ngan, F.: NOAA's HYSPLIT Atmospheric Transport and Dispersion Modeling System, Bulletin of the American Meteorological Society, 96, 2059-2077, 10.1175/bams-d-14-00110.1, 2015.

---

## Author Response (AR2)

Dear editor:

We would like to thank again the referees for their helpful comments and suggestions, which have been fully taken into account upon manuscript revision. A point-by-point response to the comments and a revised manuscript were uploaded.

Sincerely yours,

Nan Ma and co-authors

**Response to comments of referee #2**

The manuscript has been extensively edited with several new sections, resulting in a clear and comprehensive work with only some minor issues to be solved.

Response:

Thanks for the comment.

(1) The section comparing emissions with detected decrease would profit from a description of the emission data. Where are the emission data from, what are they based on etc.?

Response:

Thanks for the suggestion. The emission data used in the comparison Sect. 3.2 is from German Informative Inventory Report (UBA, 2020). This report was produced by the national co-ordination agency for the National System of Emissions Inventories (Nationales Systems Emissionsinventare; NaSE), sited within the German Federal Environmental Agency (UBA). The dataset is based on a large number of sources and publications and the detailed information is available at http://iir-de.wikidot.com/. In Sect. 3.2, we have added the following description:

"In this section, the long-term variation of total emission of major pollutants in Germany are compared with the trends of aerosol concentrations measured at the six regional background and mountain sites. The emission data is from German Informative Inventory Report (UBA, 2020) produced by the national co-ordination agency for the National System of Emissions Inventories (Nationales Systems Emissionsinventare; NaSE) sited within the German Federal Environmental Agency (UBA), and is based on a large number of sources and publications. Detailed information is available in UBA (2020)."

Reference:

German Federal Environmental Agency, UBA: German Informative Inventory Report 2020 (IIR 2020). available at http://iir-de.wikidot.com/; access: 1 March 2020.

(2) Have the differences before and after LEZ / between individual LEZ levels been tested for statistical significance? (L378+L389)

Response:

Thanks for the comment. Two-sample t-test was used to test the differences between the concentration levels before and after LEZ. And we find that all differences are statistically significant at 95 % significance level. To make it clearer, the sentences have been revised as:

"Figure 6a and b show that the eBC mass concentration and $N_{[30-200]}$ have gradually decreased after the implementation of each of new stages of LEZ. And the differences between the concentration levels at different stages are statistically significant at 95 % significance level. However, the difference between stage 2 and 3 is relatively negligible."

"Statistically significant differences are found between the concentration levels before and after the implementation of LEZ for both $N_{[30-200]}$ and eBC mass concentration. But the decrease in $N_{[30-200]}$ after 2010 (Fig. 6c) is much larger than that in eBC mass concentration (Fig. 6d)."

(3) L273 The large contribution of SOA on PM would be expected from a close relationship between decrease in precursors and decrease in PM? Since this is not the case, the sentence is confusing.

Response:

Thanks for the comment. The sentences have been revised as:

"Secondary aerosol formation processes are highly complex and nonlinear, determined not only by the concentrations of precursors but also many other factors such as solar radiation, temperature, humidity, and diffusion condition etc. Thus, the concentration of secondary aerosol might not follow the variation of precursor concentrations. It was found that secondary aerosol contributes a large fraction in particulate matters in the regional background settings (Castro et al., 1999). Therefore, discrepancies are observed between the emissions and particle concentrations although decrease trends are found in both of them."

(4) L325 It is not clear what other parameters show similar diurnal pattern as N10-30 at UB?

Response:

Thanks for the comment. The sentence has been revised as "It is interesting that at the roadside and urban background sites the trend of $N_{[10-30]}$ has similar diurnal pattern as those for $N_{[30-200]}$ and eBC mass concentration, but at regional background and low mountain range sites they look quite different."

(5) Table 5 Why there is no percentage change in temperature data if it is done for precipitation and wind speed?

Response:

Since the average ambient temperature (Celsius) may be close to 0° or below 0° in cold seasons, the calculated percentage change may be a very large number or opposite to the absolute change of temperature, which does not indicate the real trend. And absolute temperature (K) is not usually used for ambient temperature. Therefore, we decided to give only the absolute temperature change in Table 5.

(6) In figure captions, it is sometimes not clear what data are plotted; for example, in Fig. 3 the text says it is only from regional background and mountain stations but the figure caption states just Germany… The same issue is also with Fig. 10.

Response:

Thanks for the comment. The captions of Fig. 3 and Fig. 10 have been revised as:

Figure 3: Long-term change index (% of average in 2009) of measured parameters at the regional background and mountain stations sites and total emissions in Germany: (a) measured eBC mass concentration and BC emission; (b) measured $N_{[10-30]}$, $N_{[30-200]}$, $N_{[200-800]}$, and $V_{[20-800]}$, and emission of $PM_{2.5}$, $SO_2$, NMVOC, $NH_3$ and $NO_X$.

Figure 10: Annual average eBC mass concentration and $N_{[200-800]}$ measured at the regional background and low mountain range sites for the polluted and clean air mass categories. Green bars show the frequency of polluted air masses in each year.

(7) In Fig. 7, maybe ΔeBC etc. would be a better y-axis label instead of only eBC? Similarly, the same applies on b and c parts as well.

Response:

Thanks for the suggestion. Figure 7 has been modified, as shown below. Accordingly, the description of Fig. 7 in the text has been revised as: "Figure 7 illustrates the annually averaged diurnal cycles of the increments of eBC mass concentration (ΔeBC), $N_{[30-200]}$ ($\Delta N_{[30-200]}$) and $N_{[200-800]}$ ($\Delta N_{[200-800]}$). Before and after the implementation of LEZ (2010 and 2011), the increments show a sudden decrease of up to 40 % during daytime. The average ΔeBC, $\Delta N_{[30-200]}$ and $\Delta N_{[200-800]}$ during working hours (06:00 to 18:00 local time) in 2010 are respectively about 1.63, 1.33, and 1.58 times higher than those in 2011."

[Figure]

Figure 7: Average diurnal cycles of the increment of eBC mass concentration (ΔeBC, panel a), $N_{[30-200]}$ ($\Delta N_{[30-200]}$, panel b) and $N_{[200-800]}$ ($\Delta N_{[200-800]}$, panel c).

(8) In Fig. 4, the RS, UB etc. abbreviations could be explained in the caption?

Response:

Thanks for the suggestion. The caption of Fig. 4 has been revised as:

[revised manuscript text omitted]